# A high-affinity split-HaloTag for live-cell protein labeling

Yin-Hsi Lin [1,2], Julian Kompa [1], De-en Sun [1], Runyu Mao[1], Birgit Koch [1], Konstantin Hinnah[1], Jonas Wilhelm [1], Natascha Franz[1], Stefanie Kühn [1], Tanja Menche [1], Abdinasir Adow[1], Paula Breuer [1], Julien Hiblot [1] ✉ & Kai Johnsson [1,3] ✉

We introduce a high-affinity split-HaloTag comprised of a short peptide tag (Hpep, 14 residues) and a large, inactive fragment (cpHaloΔ3). Hpep binds to cpHaloΔ3 spontaneously with nanomolar affinity, enabling subsequent labeling with fluorescent HaloTag ligands. The small size of Hpep facilitates cloning-free endogenous protein tagging using CRISPR/Cas9 and the complementation of Hpep-tagged proteins can be achieved in live cells through co-expression with cpHaloΔ3 and in fixed cells through incubation with cpHaloΔ3. The approach is compatible with advanced microscopy techniques such as expansion microscopy and live-cell STED imaging. Additionally, variants of Hpep that modulate the spectral properties of labeled fluorophores enable simultaneous imaging of two different Hpep-tagged proteins via fluorescence lifetime microscopy. In summary, our high-affinity split-HaloTag is a robust and versatile tool for live-cell imaging and diverse applications in chemical biology.

The self-labeling protein HaloTag7[1], commonly known as HaloTag, has become a popular tool for live-cell fluorescence imaging, in particular in combination with super-resolution microscopy (SRM) techniques[2–4]. HaloTag reacts selectively and irreversibly with chloroalkane (CA) derivatives. Among the substrates used for fluorescence labeling, CA-conjugated rhodamines[5] stand out due to their availability in multiple colors, excellent cell permeability, high brightness, and photostability. In some cases, they also offer the advantage of fluorogenicity. Particularly in the red and near-infrared channels, HaloTag labeling provides advantages over fluorescent proteins (FPs) not only in terms of brightness, but also enables flexibility in color selection without the need for additional cloning. However, one of the main drawbacks of HaloTag is its relatively large size (33 kDa), which can potentially interfere with the folding and function of certain proteins of interest (POIs)[6].

In contrast, peptide tags (typically <20 amino acids) reduce the risk of disrupting POI functions[7–9] and often provide greater flexibility in tagging sites[10]. They have also enabled signal amplification through the use of peptide arrays[11]. Furthermore, when combined with CRISPR/Cas9 technology, short peptide tags enable more efficient and scalable generation of endogenously tagged cell lines, allowing for the study of POI functions at near-physiological levels[12]. As peptide tags are typically not intrinsically fluorescent, they require additional labeling strategies for detection. These strategies can be specific binding to fluorescent molecules (e.g., tetracysteine tag[13,14], His-tag[15,16], coiled-coil tag-probe[17–20], SunTag[11], MoonTag[21], ALFA-tag[22]) or enzyme-mediated labeling methods (e.g., Sortase A/LPXTG motif[23], BirA/acceptor peptide tag[24,25]). However, their use in live-cell imaging of intracellular proteins remains limited due to (i) cytotoxicity[13,14,24,25], (ii) low cell permeability of the substrate/probe[17–20,23–25], and/or (iii) challenges in achieving a sufficiently high signal-to-background ratio (SBR). Although some of these systems perform well for overexpressed proteins[11,13–25], for application to low-abundance, endogenously expressed proteins, the SBR remains a key challenge when only a single copy of the tag is fused.

[1]Department of Chemical Biology, Max Planck Institute for Medical Research, Heidelberg, Germany. [2]Institute of Bioengineering (BioE), École Polytechnique Fédérale de Lausanne (EPFL), Lausanne, Switzerland. [3]Institute of Chemical Sciences and Engineering (ISIC), École Polytechnique Fédérale de Lausanne (EPFL), Lausanne, Switzerland. ✉e-mail: julien.hiblot@mr.mpg.de; johnsson@mr.mpg.de

An alternative strategy to overcome these limitations involves asymmetrically splitting FPs or luciferases to create small peptide tags, such as $FP_{11}$[26] and HiBiT[27]. These small peptides spontaneously complement their corresponding large fragments, $FP_{1-10}$ and LgBiT, respectively, enabling protein visualization in living cells with minimal background[12,28,29]. However, except for mNeonGreen2$_{1-10/11}$, existing split FPs and split luciferase systems are limited in brightness and photostability[30,31], which restrict their use in SRM, especially for imaging POIs with low endogenous expression. On the other hand, split-HaloTag could harness bright, photostable fluorophores to overcome limitations of split FPs and luciferases[32]; yet, existing versions[33–36] are either not based on short peptides or lack high intrinsic affinity, limiting their use for endogenous protein tagging.

Here, we introduce a high-affinity split-HaloTag system composed of a small peptide and its complementary large protein fragment, designed as a versatile peptide tagging tool suitable for, but not restricted to, live-cell imaging applications. Demonstrated applications of the system include labeling of fixed samples, expansion microscopy (ExM), endogenous tagging via CRISPR/Cas9, live-cell stimulated emission depletion (STED) imaging, and multiplexing with fluorescence lifetime imaging (FLIM).

## Results

### Engineering high-affinity split-HaloTag pairs

We based the design of our peptide tagging tool on our previously reported split-HaloTag system[36] (Fig. 1a), which consists of a small peptide (Hpep) and a large protein fragment (cpHaloΔ). We aimed to optimize three properties of the split-HaloTag system for peptide tagging: (1) enhancing the affinity between the split fragments, (2) minimizing residual labeling activity of cpHaloΔ in the absence of Hpep, and (3) ensuring efficient labeling after reconstitution of active HaloTag from the two fragments.

We first generated a small peptide with increased affinity, followed by optimization of the large protein fragment. Throughout the

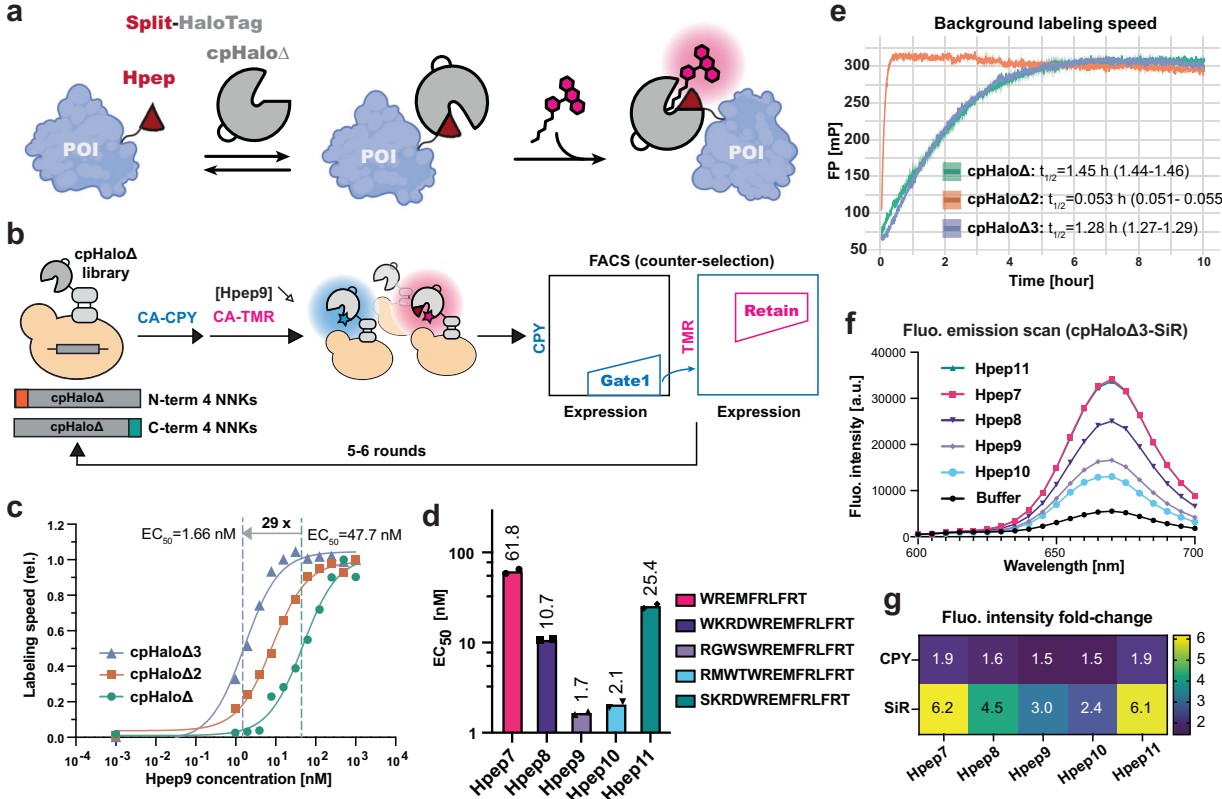

**Fig. 1 | Split-HaloTag engineering and characterization. a** A schematic illustration of the concept of protein labeling using the spontaneous self-complementing peptide-protein split-HaloTag system. POI: protein of interest. Illustration was created using Adobe Illustrator and elements exported from BioRender.com. **b** A scheme of cpHaloΔ engineering via yeast surface display. Two libraries of terminal extended cpHaloΔ variants were displayed on the yeast surface, followed by a pulse-chase labeling procedure with CPY labeling in the absence of Hpep and a subsequent incubation with TMR in the presence of Hpep9. All cpHaloΔ variants were displayed as N-terminal fusions with Aga2p and eUnaG2, which served as an expression reporter. Yeast cells exhibiting a low CPY/eUnaG2 ratio and a high TMR/eUnaG2 ratio were enriched through multiple screening rounds, during which the Hpep9 concentration was progressively decreased (from 2 to 0.25 μM for N-term NNK library and from 2 to 0.125 μM for C-term NNK library). **c–g** in vitro characterization of purified split-HaloTag fragments. **c** cpHaloΔ3 showed an improved binding for Hpep9 as determined by a kinetic assay. Labeling rates were measured using cpHaloΔ variants (2.5 nM) and CA-TMR (0.5 nM) across a twofold dilution series of Hpep9 (1000–0.98 nM). Labeling speeds were plotted against Hpep9 concentration and a sigmoidal model was fitted to the data to determine $EC_{50}$ values. Data

represent two technical replicates. $EC_{50}$ values were found to be 47.65 nM (31.28-72.51) for cpHaloΔ, 8.217 nM (6.565-10.31) for cpHaloΔ2, and 1.654 nM (1.199-2.295) for cpHaloΔ3, with the corresponding 95% confidence intervals in brackets. **d** $EC_{50}$ values of cpHaloΔ3 for five Hpep variants 7–11 determined with kinetics assay as described in (**c**) and in Supplementary Fig. 3e. Twofold dilution series from 15 μM to 0.92 nM were prepared for measurements with Hpep7, 8, and 11. For Hpep9 and 10, dilution series ranged from 1000 nM to 0.98 nM. Measurements were done with two technical replicates. **e** Background labeling activity of three cpHaloΔ variants [12.5 μM] with CA-TMR [50 nM] in the absence of Hpep, measured by fluorescence polarization (FP). A one-phase association model was fitted to each dataset to determine the labeling half-time ($t_{1/2}$), shown next to each cpHaloΔ variant along with the corresponding 95% confidence intervals (in brackets). Representative data from duplicate measurements. **f** Fluorescence emission profiles of SiR-labeled cpHaloΔ3 [100 nM] in the absence and presence of synthetic Hpep variants 7–11 [500 nM] after 5-h incubation. Data present mean values from triplicate measurements. **g** Fluorescence intensity increases (average values from triplicate measurements) are represented as fold-changes upon binding to Hpep variants, compared to CPY or SiR-labeled cpHaloΔ3 in the absence of Hpep.

engineering process, the $EC_{50}$ value, defined as the Hpep concentration required to achieve half-maximum labeling speed of cpHaloΔ, was used as a proxy for binding affinity. To select Hpeps that bind to cpHaloΔ with lower $EC_{50}$, we incorporated beneficial mutations from previously reported Hpep sequences (Supplementary Table 1)[37]. This resulted in the three peptides Hpep9–11, with $EC_{50}$ values ranging from 40 to 235 nM. Subsequently, we engineered cpHaloΔ for enhanced binding to the highest affinity peptide, Hpep9, using yeast surface display (YSD) in combination with fluorescence-activated cell sorting (FACS). We constructed two separate libraries by adding a stretch of four randomized residues to either the N or C-terminus of cpHaloΔ. The rationale behind the construction of these N-terminal and C-terminal extensions was to create additional interactions with Hpep9. To identify cpHaloΔ variants with low residual labeling activity in the absence of Hpep but robust labeling in the presence of Hpep9 (which correlates with binding affinity), we established a pulse-chase labeling procedure (Fig. 1b). Yeast libraries were first labeled with CA-CPY (CA derivative of carbopyronine) without Hpep9, followed by labeling with a spectrally distinct dye, CA-TMR (CA derivative of tetramethylrhodamine), in the presence of Hpep9. The labeling signal observed under these conditions was used as a qualitative measure reflecting the combined effects of higher binding affinity and faster labeling kinetics. By gradually decreasing Hpep9 concentration and incubation time, cpHaloΔ variants yielding stronger CA-TMR labeling signals were selectively enriched over iterative rounds (Fig. 1b and Supplementary Figs. 1 and 2). Next-generation sequencing analysis facilitated the identification of strongly enriched cpHaloΔ variants (Supplementary Fig. 1b–e), which were further characterized as purified proteins. Among 24 variants, 22 variants showed enhanced binding to Hpep9 (Supplementary Fig. 3a, b). We then combined the best-performing terminal extension sequences with the stabilizing mutations (E20S, N119H, V184E, V197K, and the redesigned circular permutation linker), previously identified for the improved variant of cpHaloΔ2[38] (Supplementary Fig. 3c). This resulted in the final variant, cpHaloΔ3.

In comparison to cpHaloΔ and cpHaloΔ2, cpHaloΔ3 exhibited a 29-fold and fivefold lower $EC_{50}$ for Hpep9, respectively (Fig. 1c and Supplementary Fig. 3c, d). Relative to cpHaloΔ2, cpHaloΔ3 also showed an improved binding to the previously reported peptides Hpep7 and 8, and for the new peptides Hpep10 and 11, with $EC_{50}$ values ranging from single-digit to double-digit nanomolar concentrations (Fig. 1d and Supplementary Fig. 3e). Regarding labeling rates with CA-TMR at saturating Hpep9 concentrations, cpHaloΔ3 showed an about 1.4-fold faster labeling rate than cpHaloΔ, and a similar labeling rate as cpHaloΔ2 (1.1-fold slower). Furthermore, cpHaloΔ3 showed a similar spontaneous labeling rate in the absence of Hpep as cpHaloΔ, but a 25-fold slower residual labeling rate than cpHaloΔ2 (Fig. 1e), which resulted in lower background labeling signal in cellular assays compared to cpHaloΔ2 (Supplementary Fig. 4). We further confirmed that the reduced residual activity of cpHaloΔ3 is not due to a decrease in thermostability, as its melting temperature was identical to that of cpHaloΔ2 at 46.3 °C (Supplementary Fig. 5). The low residual labeling rate of cpHaloΔ3, together with its rapid labeling in the presence of Hpep and enhanced binding across a series of Hpep variants, makes it an attractive candidate for peptide tagging.

The crystal structure of TMR-labeled HaloTag (PDB ID: 6Y7A) indicates that Hpep contributes to the formation of the HaloTag ligand-binding pocket (Supplementary Fig. 6a). We hypothesized that mutations in Hpep could influence the fluorescence intensity (FI) of fluorophore-labeled cpHaloΔ3. Therefore, we measured the FI of CPY or SiR-labeled cpHaloΔ2 and cpHaloΔ3 in the presence or absence of different Hpep variants (Hpep7–11). Upon binding to different Hpep variants, cpHaloΔ3 showed a 1.5 to 6.2-fold increase in FI. The fluorescence "turn-on" upon Hpep binding was more pronounced with cpHaloΔ3 than with cpHaloΔ2, and with SiR (silicon-rhodamine)

compared to CPY (Fig. 1f, g and Supplementary Fig. 6b, c). Among all tested Hpeps, Hpep7 and Hpep11 showed the most pronounced turn-on. The brightest split-HaloTag pairs (Hpep7 and Hpep11 in combination with cpHaloΔ3) for SiR even showed a slightly enhanced FI (~20% increase) relative to full-length HaloTag (Supplementary Fig. 6d, e).

## Biophysical characterization of split-HaloTag pairs

To directly assess the binding interaction between Hpep and cpHaloΔ, biophysical measurements were performed to complement the labeling reaction-based $EC_{50}$ analyses described above. The two peptides Hpep9 ($EC_{50}$ = 1.7 nM), which exhibited the lowest $EC_{50}$ for cpHaloΔ3, and Hpep11 ($EC_{50}$ = 25.4 nM), which led to the brightest overall signal for the complemented split-HaloTag, were selected for detailed analysis with cpHaloΔ2 and cpHaloΔ3.

First, we determined the dissociation constant ($K_d$) using chemically synthesized TMR-Hpep conjugates and fluorescence polarization measurements. cpHaloΔ3 showed high affinities for both Hpep9 ($K_d$ = 6.5 nM) and Hpep11 ($K_d$ = 13.5 nM), corresponding to 4 and 2.8-fold tighter binding, respectively, compared with cpHaloΔ2 (Supplementary Fig. 7a). To investigate the molecular basis underlying the better binding properties of cpHaloΔ3 for these two Hpeps compared to cpHaloΔ2, we generated structural models using AlphaFold3 (AF3)[39] to identify potential interactions involving the terminal extensions of cpHaloΔ3 (Supplementary Fig. 8). Due to low confidence scores (pLDDT) in these regions, potentially indicative of intrinsic flexibility, reliable interaction mapping was precluded. We therefore performed molecular dynamic (MD) simulations based on the AF3 predicted models to assess the stability and dynamics of the complexes, focusing our analysis on the structural integrity of Hpep within each complex to evaluate the influence of cpHaloΔ variants on peptide stability. Given the low confidence in the AF3-predicted structure of Hpep9, we focused our analysis on Hpep11 (Supplementary Fig. 7b). In the cpHaloΔ2/Hpep11 complex, Hpep11 exhibited pronounced structural fluctuations, as reflected by increased root-mean-square deviation (RMSD) and a significant structural distortion after 50 nanoseconds (ns) of simulation (Supplementary Fig. 7c and Supplementary Videos 1 and 2). In contrast, the cpHaloΔ3/Hpep11 complex showed markedly reduced fluctuations, with Hpep11 maintaining a more stable conformation throughout at least 100 ns of simulation. The result highlights the role of the engineered terminal extension (GDVE) on cpHaloΔ3 in stabilizing the interaction complex with Hpep11, through its engagement with the N-terminal α-helix of Hpep11.

For imaging applications using Hpep as a labeling tag, strong binding between the labeled-cpHaloΔ and Hpep is essential, as dissociation of the labeled cpHaloΔ can lead to nonspecific signals. To evaluate the impact of fluorescent labeling on binding affinity, we then measured the dissociation constant ($K_d$) of SiR-labeled cpHaloΔ3 and compared it to that of the non-labeled cpHaloΔ3 (Supplementary Fig. 7d), using TMR-Hpep conjugates. Both Hpep9 and Hpep11 showed significantly higher affinity for SiR-labeled cpHaloΔ3, with $K_d$ values below 5 nM (Supplementary Fig. 7a and d). Notably, the affinity enhancement was more pronounced for Hpep11 (tenfold) than for Hpep9 (twofold), resulting in a lower $K_d$ for Hpep11. To assess whether fluorophore conjugation on the peptide affects affinity measurements and to measure binding kinetics, we then performed bio-layer interferometry (BLI) using biotin-Hpep conjugates (Supplementary Fig. 7e, f and Supplementary Table 2). The strongest interaction was observed between cpHaloΔ3-SiR and Hpep11, with a $K_d$ of 4.3 nM and a slowest dissociation rate constant of $k_{off} \approx 2.14 \times 10^{-4}$ s$^{-1}$. The consistent $K_d$ values across these two methods, fluorescence polarization (FP) assay using fluorophore-Hpep11 conjugate and BLI using biotin-Hpep11 conjugate indicated that fluorophore conjugation on the peptide has minimal impact on the measured binding affinity (Supplementary Table 22).

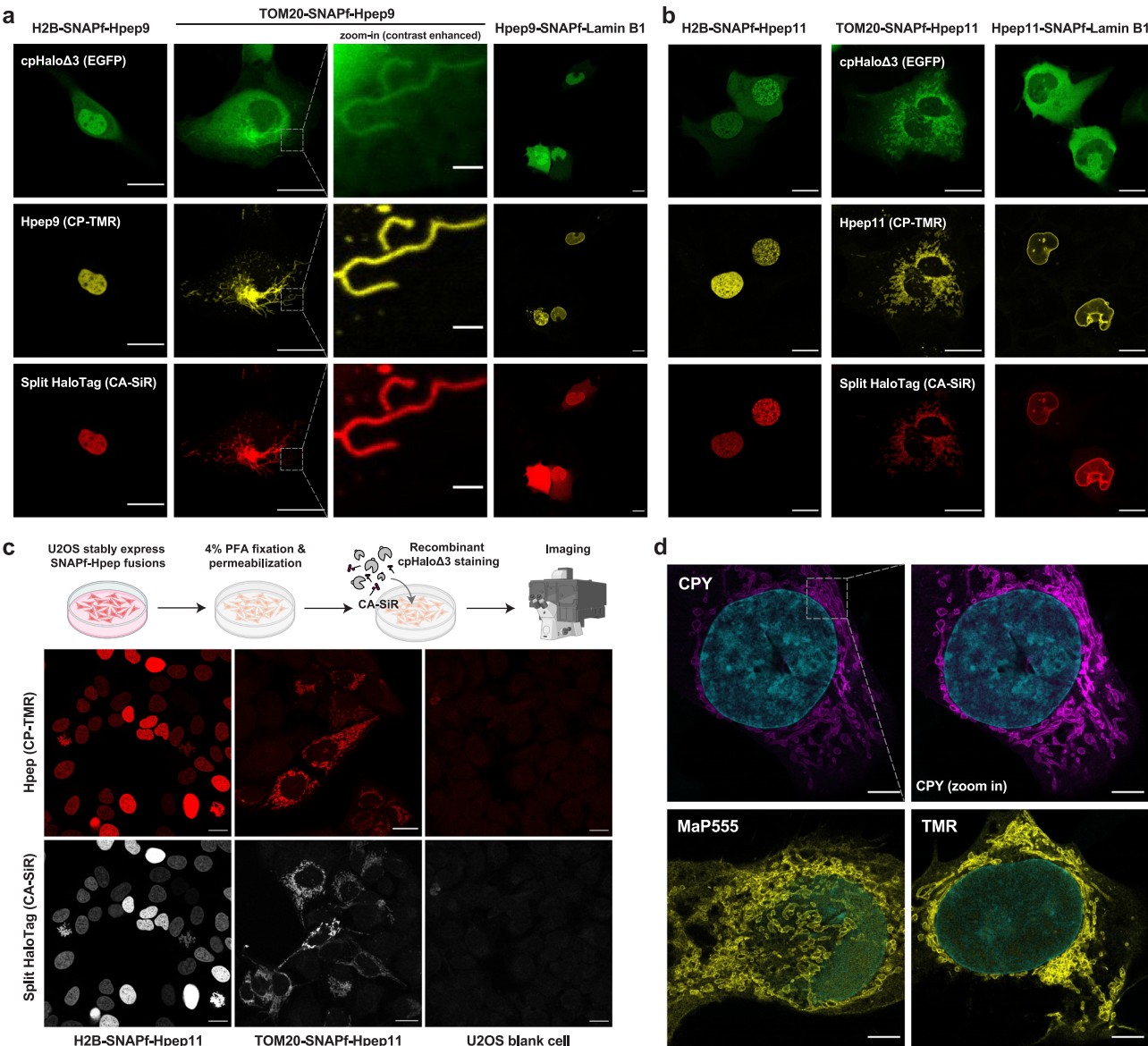

**Fig. 2 | Protein labeling and imaging in mammalian cells using Hpep.**
**a**, **b** Confocal live-cell imaging of U2OS cells co-expressing both split fragments, revealing the spontaneous complementation of EGFP-cpHaloΔ3 with SNAPf-fused Hpep9 (**a**) or Hpep11 (**b**) at their specific subcellular localizations. SNAPf refers to SNAP^E30R, a faster variant of SNAP-tag. Cells were labeled with SNAPf substrate CP-TMR [250 nM] and HaloTag substrate CA-SiR [100 nM] for one hour. Scale bar: 20 μm and 2 μm for the zoom-in images. Representative images were obtained from at least five independent biological experiments, with more than 10 cells analyzed per replicate. **c** Post-fixation labeling of Hpep11-tagged targets (H2B and TOM20) with recombinant cpHaloΔ3 [1 μM], CA-SiR [500 nM], and CP-TMR [500 nM] after overnight incubation at 4 °C. Scale bar: 20 μm. Representative

images were obtained from two independent biological experiments, analyzing over five fields of view per replicate. Illustration was created using Adobe Illustrator with elements originally created with BioRender.com. **d** Representative confocal images of U2OS cells co-expressing EGFP-cpHaloΔ3 and TOM20-SNAPf-Hpep11 after fixation and expansion. The labeling with HaloTag CA-ligands [100 nM] was performed before fixation for 2 h. Yellow: CA-MaP555 and CA-TMR labeling. Magenta: CA-CPY labeling. Cyan: Hoechst 33342 nucleic acid stain. Scale bar: 10 μm (post-expansion). Representative images were collected from more than three independent biological replicates, each comprising analyses of more than three fields of view.

## Live-cell imaging of Hpep-tagged targets

For labeling in mammalian cells, we used cpHaloΔ3 in combination with Hpep9 and Hpep11. Hpep was fused either to the histone H2B (nucleus), the mitochondrial import receptor subunit TOM20 (outer mitochondrial membrane), or the laminin subunit beta 1 (nuclear envelope). To visualize cpHaloΔ3 and Hpep9 or Hpep11 fusions, the Hpep fusions also included SNAPf (SNAP^E30R), and cpHaloΔ3 was fused to EGFP. Staining U2OS cells co-expressing the two split-HaloTag components with SNAPf substrate CP-TMR revealed the expected localization of Hpep and the successful recruitment of cpHaloΔ3. The spontaneous reconstitution of the split-HaloTag enabled specific

labeling of the desired sub-cellular structures with CA-SiR, while the unbound cpHaloΔ3 did not produce significant unspecific SiR signal, except in the case where Lamin B1 was tagged with Hpep9 (Fig. 2a, b). In general, Hpep11 fusions led to a higher signal-to-background ratio (SBR) in the SiR channel than Hpep9, and the difference was most pronounced in the case of laminin (Fig. 2a, b and Supplementary Fig. 9a–d). No SiR labeling was observed in cells expressing only one of the two split-HaloTag fragments under same image acquisition settings (Supplementary Fig. 10a). To rule out any potential influence of EGFP on the performance of split-HaloTag in these experiments, we generated a construct with a self-cleaving T2A sequence between

cpHaloΔ3 and EGFP. cpHaloΔ3 alone demonstrated similar performance as EGFP-cpHaloΔ3 fusion protein (Supplementary Fig. 10b).

To evaluate the performance of Hpep11 at different insertion sites, we generated EGFP fusion constructs with Hpep11 placed at either terminus or within an internal loop region[40]. These constructs were expressed in either U2OS or HeLa cells and analyzed by flow cytometry (Supplementary Fig. 11a, b). At both termini, Hpep11 enabled efficient and comparable cpHaloΔ3 labeling, resulting in a strong increase in SiR signal relative to the negative control lacking Hpep. Notably, although Hpep was not designed for loop insertion, it still allowed specific cpHaloΔ3 labeling when inserted at a loop region, as confirmed by live-cell imaging (Supplementary Fig. 11c). This was accompanied by a moderate reduction in EGFP fluorescence, suggesting some influence on EGFP folding (Supplementary Fig. 11a, b).

### Post-fixation imaging Hpep-tagged targets

Labeling of Hpep11-tagged proteins for post-fixation imaging can be achieved in two ways: either by staining cells co-expressing cpHaloΔ3 (Supplementary Fig. 10c), or by incubating Hpep11-expressing cells with recombinant cpHaloΔ3 protein and a HaloTag ligand (CA-SiR or CA-CPY) after cell fixation and permeabilization. To assess the abundance and the localization of Hpep11, a SNAPf ligand (CP-TMR) was also included. The signal intensities in both channels (SNAPf-Hpep and cpHaloΔ3) were well-correlated and no detectable signal was observed in blank U2OS cells, confirming the specific labeling of the Hpep11-tagged subcellular structures (Fig. 2c and Supplementary Figs. 9e, f and 10d).

The ability to image cells after fixation prompted us to explore the compatibility of our tagging system with expansion microscopy (ExM), a powerful super-resolution technique that enables imaging of target proteins beyond the diffraction limit using a conventional light microscope[41]. We imaged cells co-expressing TOM20-Hpep11 and cpHaloΔ3, which were labeled with CA-CPY, CA-TMR, or CA-MaP555. After expansion, ExM enabled clear distinction of mitochondrial membrane, where Hpep11-tagged TOM20 localized (Fig. 2d), offering a sharp contrast with the lumen, which was challenging to resolve using traditional confocal imaging.

### Hpep11 allows cloning-free CRISPR/Cas9 genome editing

Endogenous tagging allows protein visualization with minimized risk of artifacts typically associated with transgene over-expression. Owing to its small size of 14 residues, Hpep facilitates genome editing through CRISPR/Cas9 in a cloning-free manner. To demonstrate the use of split-HaloTag for imaging endogenously tagged proteins, six cellular targets were selected, with the corresponding gene names indicated in parentheses: histone H2B type 2-E (*HIST2H2BE*), translocase of outer mitochondrial membrane 20 (*TOMM20*), lamin A/C (*LMNA*), clathrin light chain A (*CLTA*), Sec61 subunit beta (*Sec61B*), and vimentin (*VIM*). These proteins, which possess different subcellular localizations and a wide range of expression levels were selected for either N or C-terminal tagging (Supplementary Table 3). Guided by insights from previous studies[12,29], Hpep was introduced into the genomic loci of the aforementioned targets in U2OS cells via a ribonucleoprotein (RNP)-mediated CRISPR approach, with synthetic single-stranded oligonucleotides (ssODN) as donor DNA templates (Fig. 3a). To demonstrate adaptability to diverse experimental workflows, the cpHaloΔ3-expressing cassette was introduced via transient transfection, stable integration using the Flp-In T-REx system, or AAVS1 safe harbor targeting[42]. Upon labeling with CA-SiR, cells with Hpep integration displayed a distinct population compared to cpHaloΔ3-only cells, enabling FACS-based enrichment (Fig. 3b). Subsequent labeling of these cell lines with CA-CPY or CA-SiR revealed the proper localization of all Hpep11-tagged targets, as observed with confocal microscopy, with clear distinction from background fluorescence (Fig. 3b and Supplementary Fig. 12a).

Through tagging TOM20 endogenously with each of the five Hpep variants (Hpep7–11), we confirmed that Hpep11 outperformed the others, exhibiting the highest SBR in both microscopy and flow cytometry analysis (Fig. 3c, d and Supplementary Fig. 12b). Moreover, Hpep11 showed a comparable performance to full-length HaloTag. Similar results were obtained in another cell line, 293FT cells (Supplementary Fig. 12c), with Hpep11 integrated at the genomic locus of nucleolar protein 10 (*NOP10*). We further evaluated seven additional CA-fluorophores (TMR, MaP555, MaP618, and Janelia Fluor dyes JFX608, JF635, JF646, and JF669), spanning the yellow-green to far-red spectrum, for staining endogenously Hpep11-tagged U2OS cell lines (Supplementary Fig. 12d, e). Signal intensity and SBR were assessed by confocal microscopy for two targets, H2B and lamin A/C. All dyes enabled clear visualization of H2B, whereas MaP618 and JF635 yielded weak fluorescence for lamin A/C.

### Hpep11 for live-cell STED imaging of endogenously tagged targets

Live-cell STED imaging is a powerful method to study organelles or subcellular protein localizations below the diffraction limit. However, it is demanding on the spectral properties of the fluorescent labels, and synthetic fluorophores generally outperform FPs in STED imaging[43,44]. To investigate whether split-HaloTag meets the criteria for live-cell STED imaging, we imaged U2OS cells expressing Hpep11-tagged TOM20, either overexpressed or endogenously tagged (Fig. 4a). In both cases, Hpep11 tagging allowed live-cell STED imaging of TOM20 labeled with SiR, which is one of the most widely used dyes for STED imaging[45]. The resolution in STED was markedly improved in comparison to that achieved by confocal laser scanning microscopy (CLSM, Fig. 4b). Split-HaloTag demonstrated comparable performance to that of full-length HaloTag under STED imaging (Supplementary Fig. 13), revealing hollow mitochondrial tubules and a spot-like distribution of TOM20 proteins decorating the mitochondrial membrane[46,47].

Subsequently, we performed STED imaging on two additional targets. Tagging the corresponding genes, *CLTA* and *TUBB4B* (gene encoding for Clathrin light chain A and Tubulin beta 4B), with Hpep11 in cpHaloΔ3-expressing cells enabled the visualization of clathrin-coated pits and microtubules below the diffraction limit (Fig. 4d, e). The microtubule diameter is significantly decreased using STED microscopy compared to diffraction-limited confocal imaging and we obtained a full-width at half-maximum (FWHM) of approximately $84 \pm 26$ nm (Fig. 4f), and these values are in good agreement with previously reported apparent microtubule width of 40–80 nm[48,49].

### Live-cell FLIM imaging of Hpep-tagged targets

In addition to intensity-based imaging, fluorescence lifetime is another powerful approach for creating contrast. Differences in fluorescence lifetime can be used for the elimination of background fluorescence, such as autofluorescence[50], and the discrimination between spectrally overlapping fluorophores. As the fluorescence intensity of SiR-labeled cpHaloΔ3 is influenced by binding to Hpep8–11, we determined the fluorescence lifetime values of SiR-labeled cpHaloΔ3 in the absence and presence of each of the four Hpeps as protein fusions to H2B in living U2OS cells. SiR-labeled cpHaloΔ3 alone exhibited a fluorescence lifetime of 2.3 or 2.5 ns, depending on its subcellular localization (Fig. 5a). In contrast, labeled split-HaloTag reconstituted with Hpep8–11 exhibited a fluorescence lifetime ranging from 2.2 to 3.5 nanoseconds (Fig. 5b and Supplementary Table 4). For Hpep11, the difference in fluorescence lifetime of labeled cpHaloΔ3 in the absence and presence of Hpep11 was larger than 1 ns. This difference in fluorescence lifetimes can be exploited to separate potential background signal from free cpHaloΔ3 and on-target signal from reconstituted split-HaloTag in live-cell FLIM imaging using phasor plot analysis. This was demonstrated for both Lamin B1-overexpressing and TOM20

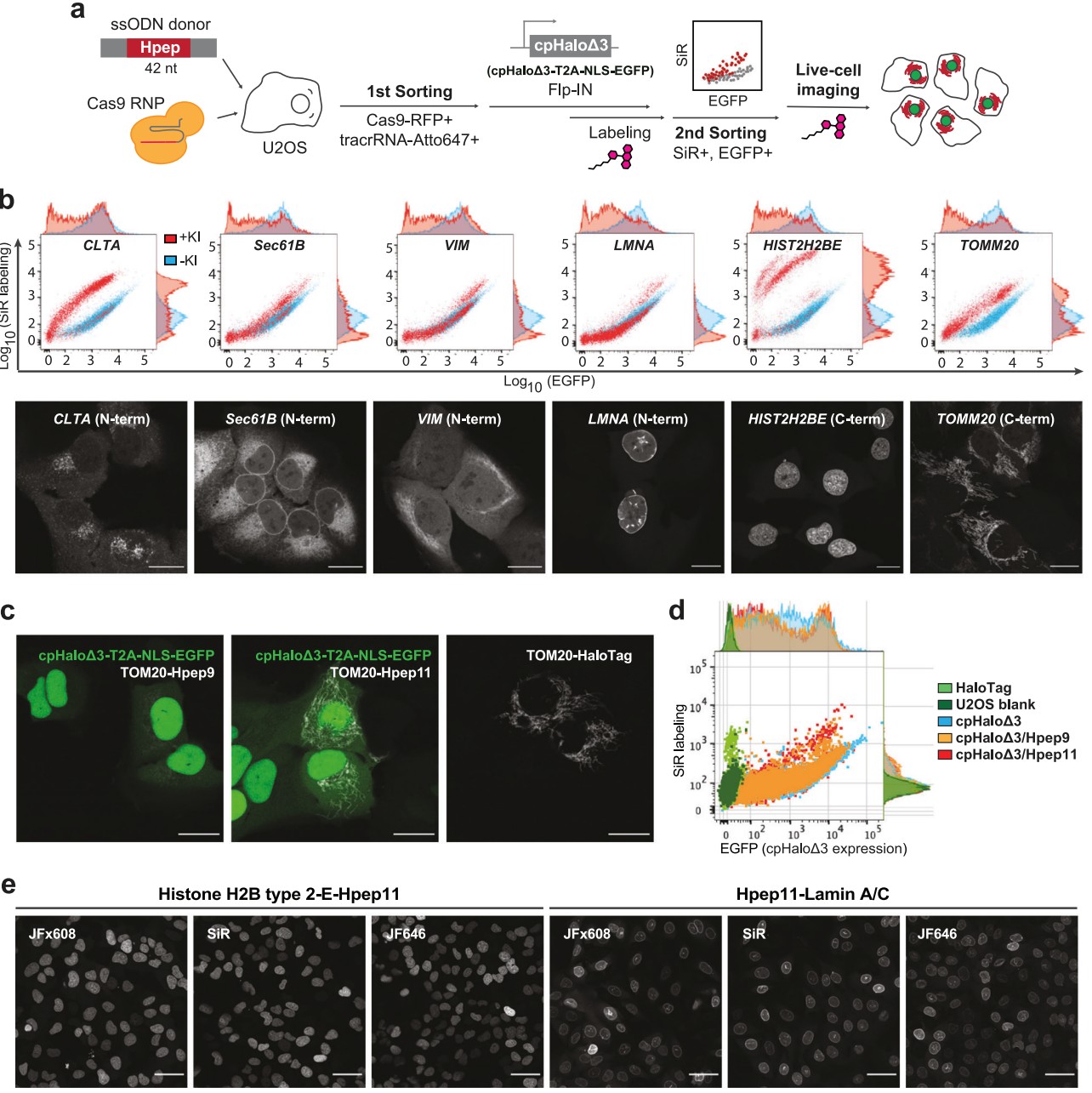

**Fig. 3 | Endogenous protein tagging with Hpeps and live-cell imaging. a** A schematic illustration of Hpep knock-in (KI) cell line workflow via CRISPR/Cas9 RNP approach (some elements adapted from BioRender.com). Two rounds of FACS sorting were performed: 1st round aiming to enrich transfected cells (double-positive for Cas9-RFP and tracrRNA-Atto647), and 2nd round for the enrichment of Hpep KI cells showing increased CA-SiR labeling signal. **b** Flow cytometry analysis (upper) and representative live-cell confocal images (lower row) of the six endogenous Hpep11 tagging cell lines after second cell sorting. U2OS cells stably expressing cpHaloΔ3-T2A-NLS-EGFP (−KI) and the Hpep11 tagging cell lines were labeled one hour with CA-SiR [100 nM] for flow cytometry experiment and one hour with CA-CPY [100 nM] for imaging. Images were taken with optimized laser settings for each target. Scale bar: 20 μm. **c, d** Labeling of TOM20 endogenously tagged with either intact HaloTag or two split-HaloTags and analysis via confocal live-cell imaging (**c**) and flow cytometry (**d**). Cells were labeled with CA-SiR [100 nM] for one hour. Scale bar: 20 μm. **e** Live-cell imaging of the CRISPR cells with the indicated CA-fluophores [100 nM] after two-hours labeling. The entire dataset can be found in Supplementary Fig. 12. Scale bar: 50 μm. Images in (**a**–**c**) are representative of at least three independent biological experiments, each including more than three fields of view or at least 20 cells. Images in (**e**) and Supplementary Fig. 12 were acquired from a single experiment, with at least three fields of view analyzed for each staining condition.

knock-in (KI) cell lines following overnight labeling with CA-SiR, extreme conditions under which background signal can be detected (Fig. 5c). Motivated by these results, the simultaneous imaging of two targets using different Hpeps through FLIM multiplexing was explored. For this, we tagged TOM20 with Hpep11 and H2B with either Hpep8, 9, or 10 (Fig. 5d). After labeling with SiR, the two structures could be well-separated by FLIM using phasor analysis for all Hpep

pairs tested. The pair of Hpep10 and Hpep11 showed the best performance when simultaneously imaging two targets, due to their large differences in fluorescence lifetime.

## Discussion
Although a variety of split-HaloTag systems have been reported previously[32–35], none of them undergo spontaneous reconstitution at

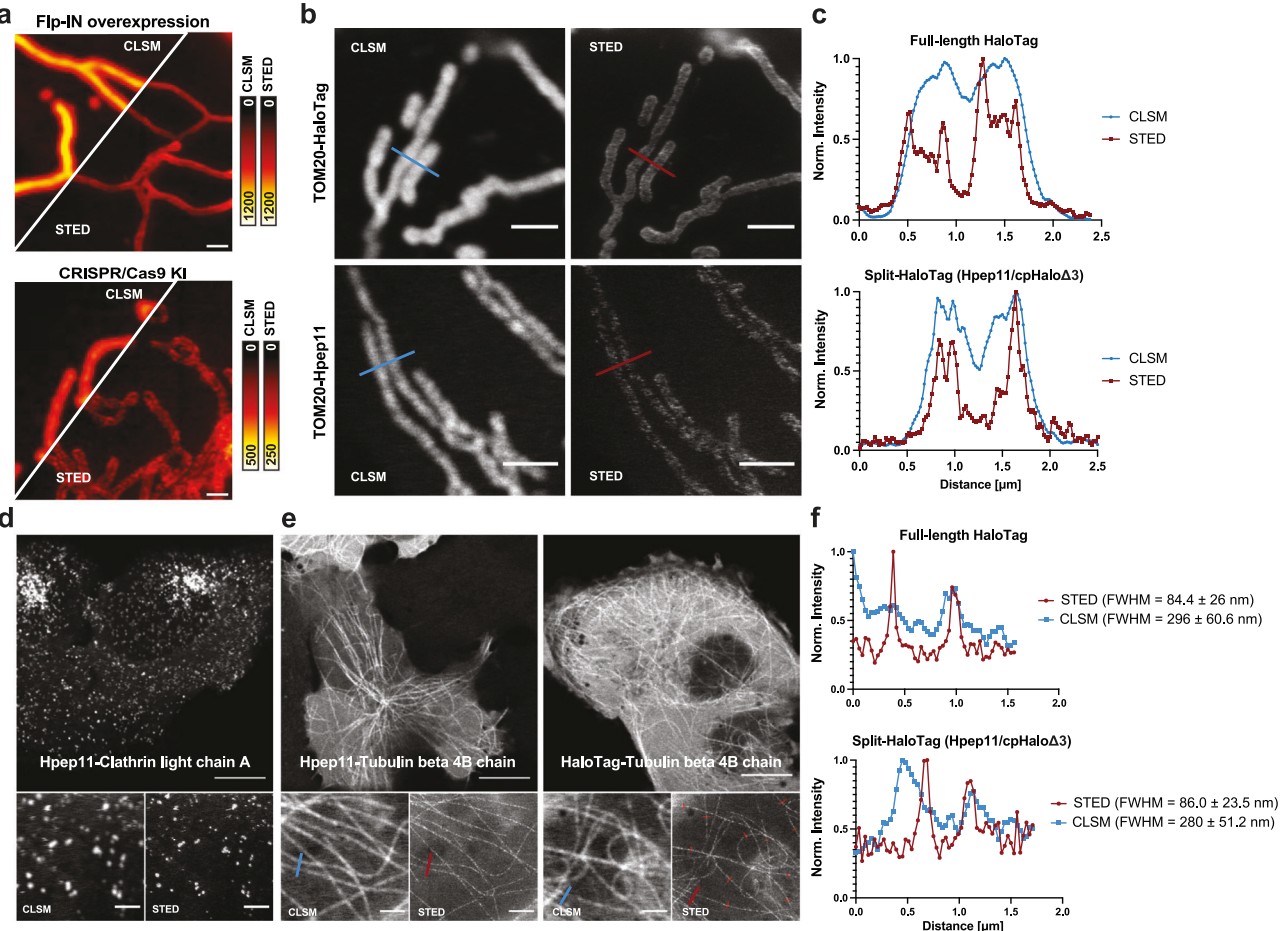

**Fig. 4 | Live-cell STED imaging of the Hpep11-tagged cell lines. a** Confocal laser scanning microscopy (CLSM) and STED images of mitochondria in U2OS cells co-expressing cpHaloΔ3 and TOM20-Hpep, either overexpressed or endogenously tagged. Scale bar: 1 μm. Pixel intensities scaled according to reference bar. **b** Representative CLSM, STED images of the CRISPR/Cas9 KI cells expressing TOM20 tagged with intact HaloTag (upper) or Hpep11 (bottom). For (**a**) and (**b**), images are representative of n = 3 independent experiments, with three images acquired from each experiment. **c** Intensity profiles along mitochondrial tubules (red and blue lines in **b**). Scale bar: 2 μm. **d** Representative CLSM and STED images of endogenous Hpep11-tagged clathrin with cpHaloΔ3 co-expression. Scale bar:

10 μm (overview) and 2 μm (magnification). For (**c**) and (**d**), n = 1 experiment, with three images obtained. **e** Representative CLSM, STED images of endogenously tagged tubulin beta 4B with Hpep11. Scale bar: 10 μm (overview) and 2 μm (magnification). **f** Intensity profiles along tubulin filaments (red and blue lines in **e**). Means ± s.d. of the filament diameters were calculated as full width at half maximum (FWHM) from n = 16–20 microtubule filaments, ≥ 2 images. A slight increase in cytosolic signal was noted in cells tagged with split-HaloTag at *TUBB4B*, compared to cells tagged with the full-length HaloTag, which may result from the presence of unbound but labeled cpHaloΔ3. All images were acquired after labeling with CA-SiR [100 nM] for one hour.

nanomolar concentrations of both fragments without additional protein-protein interactions. In this study, we report the development of a high-affinity split-HaloTag system composed of two split fragments, which are a 14-residue peptide (Hpep) and a 35 kDa protein fragment (cpHaloΔ). The engineering of both fragments resulted in cpHaloΔ3 and four Hpeps (Hpep8–11), interacting with $EC_{50}$ values ranging from 2 to 25 nM. All four peptides can recruit cpHaloΔ3 intracellularly to tagged POIs, enabling efficient and specific labeling with HaloTag ligands. Among them, Hpep11 yields an exceptional SBR, even under challenging conditions such as endogenous protein tagging, where Hpep11-tagged protein levels can be orders of magnitude less abundant than those achieved through CMV promoter-driven overexpression. The high SBR of cpHaloΔ3/Hpep11 pair can be attributed to multiple factors: an increased affinity of cpHaloΔ3 for Hpep11 when cpHaloΔ3 is labeled with SiR (with the $K_d$ decreasing from 13.5 to 1 nM, Supplementary Table 22), slow off-rates of cpHaloΔ3-SiR, a low residual labeling activity of cpHaloΔ3 in the absence of Hpep11, and a substantial fluorescence intensity increase of cpHaloΔ3-SiR upon binding to Hpep11 (sixfold in vitro). The cpHaloΔ3/Hpep11 pair showed comparable performance to intact HaloTag in all imaging applications tested.

The small size of Hpep offers several advantages over regular protein tags. First, the generation of Hpep CRISPR knock-in cell lines is cloning-free, making it easily scalable. It was already demonstrated for other small peptide tags, such as FP11 (16 aa) of split FPs and HiBiT (11 aa) of split NanoLuc, that these tags can be systematically introduced into the genomic loci of a library of protein targets, enabling the visualization of these targets at near-native expressions and facilitating the study of their functions[12,28,29]. In this study, we have explored different methods for introducing the cpHaloΔ3-expressing cassette into U2OS, 293FT, and HeLa cells, underscoring the versatility of this approach. Furthermore, the small Hpeps could enable the tagging of POIs that are sensitive to direct fusions to larger tags. The benefit of post-translationally attaching a large tag compared to a direct fusion has been demonstrated previously in yeast with several plasma membrane proteins[7–9].

The high-affinity split-HaloTag complements other live-cell compatible peptide tags. Epitope-nanobody/scFv pairs typically have very high-affinity (with $K_d$ values as low as 26 pM for ALFA-tag/NbALFA[16] and 30 nM for MoonTag/gp41 nanobody[21,51]), but they often suffer from high non-specific background fluorescence, as the fluorescently labeled binders remain fluorescent regardless of whether they are

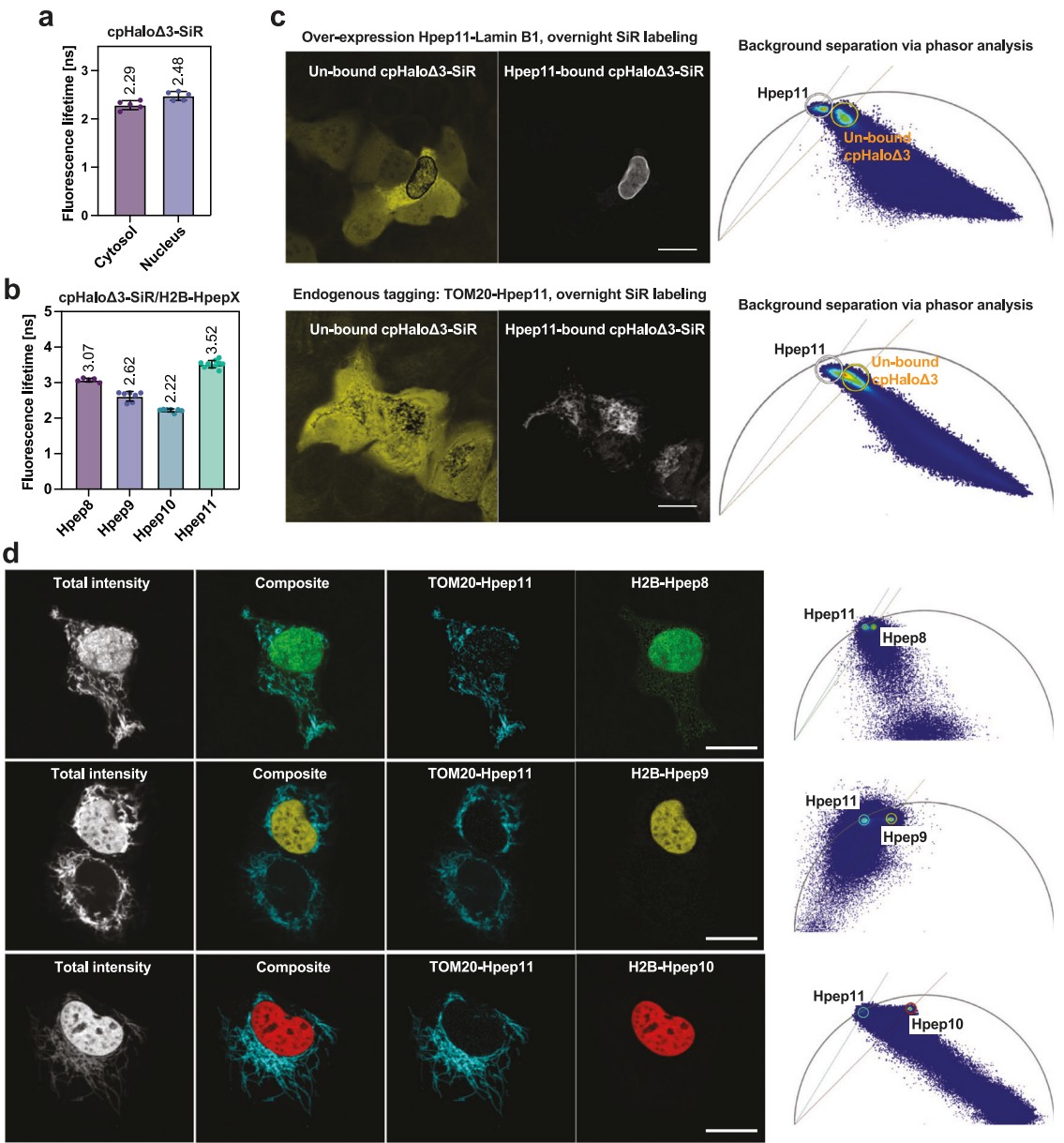

**Fig. 5 | Hpep variants for live-cell FLIM imaging. a** Fluorescence lifetimes ($\tau$) of SiR-labeled cpHaloΔ3 in cytosol and nucleus of U2OS cells. **b** Fluorescence lifetimes ($\tau$) of SiR-labeled cpHaloΔ3 bound to Hpep variants 8–11, fused to H2B. (mean fluorescence lifetimes are represented here, for respective s.d. and $N$ see Supplementary Table 4 and Source Data Excel sheet). **c** Discrimination between on-target labeling and the background signal using phasor analysis. To generate sufficient unspecific background, cells were labeled with CA-SiR [100 nM] for 20 h before imaging. The separated images (left) were generated for each cluster defined on the wavelet-filtered phasor plots (right), which represented a greater challenge for the endogenous TOM20 KI cells due to the overlapping clusters. Scale bar: 20 μm. **d** Pairwise combination of Hpep11 (TOM20-tagged) with the other three Hpep variants 8, 9, and 10 (H2B-tagged) for FLIM multiplexing. The total fluorescence intensity, composite, the two separated species, and their corresponding wavelet-filtered phasor plot used for species separation are presented. Scale bar: 20 μm. All images in (**c**) and (**d**) are representative of at least two biological replicates, with a minimum of five cells analyzed per experiment.

bound to their target epitopes or not. As a result, achieving a high SBR for low-abundance POIs often requires multiple epitope repeats[37] or careful optimization of nanobody/scFv expression[52]. In contrast, in our system, background signal can be minimized by using fluorogenic dyes such as SiR, without the need for tuning of cpHaloΔ3 expression levels, or even eliminated through fluorescence lifetime segmentation (Fig. 5c). FP$_{11}$/FP$_{1-10}$ pairs of split FPs typically do not show background fluorescence before complementation[30], but perform less well in the red channel. In contrast, our system performs well across the orange to far-red channels and enables the use of different colors without the need for additional cloning efforts (Supplementary Fig. 12b, c). By labeling with CA-fluorophores, which possess exceptional photophysical properties, our system supports super-resolution techniques,

including ExM and live-cell STED imaging. Another attractive application of our split-HaloTag is in FLIM-based multiplexing of two targets tagged with different Hpep variants, which can be labeled through co-expression of a single complementing protein (cpHaloΔ3). Labeling of two different FP$_{11}$ peptides, in contrast, requires expression of two different large fragments (FP$_{1-10}$) for specific labeling[53]. We have also demonstrated the possibility of insertion of Hpep11 at different positions within a POI, an important feature shared with several other peptide-based systems.

In summary, our high-affinity split-HaloTag system integrates the key advantages of both peptide tags and HaloTag. It combines the genome-editing convenience of peptide tags with the labeling versatility of HaloTag ligands. This system yields high SBR without the need

for specific optimization and supports multiplexed imaging using a single complementing fragment. However, the system of course also has limitations. Most importantly, as for most other peptide-based tagging systems, intracellular labeling of Hpep-tagged POIs requires co-expression of an appropriately localized cpHaloΔ[54]. However, this limitation could also have its benefits as it could be exploited for achieving cell-type- or compartment-specific labeling[55].

Beyond the applications demonstrated in this study, we foresee broader utility for our high-affinity split-HaloTag pairs, extending beyond imaging and cellular assays. For instance, the system could be combined with functional HaloTag ligands, such as HaloPROTACs[56,57] or applied in model organisms, where various HaloTag-derived tools have been already successfully employed[33,36,58–61]. Taken together, our split-HaloTag pairs should become powerful tools for live-cell imaging and beyond.

## Methods

### General information
The compositions of buffers and media used in this study are described in Supplementary Table 5. DNA purification was performed with QIAquick PCR purification kit (QIAGEN) and plasmid extraction from *E. coli* was performed using QIAprep miniprep kit (QIAGEN) according to manufacturer's procedures. DNA concentration was determined by measuring absorption at 260 nm with a NanoDrop 2000c spectrometer (Thermo Fisher Scientific). Fluorescent HaloTag and SNAPf substrates were synthesized following literature procedures[62,63] by B. Réssy and D. Schmidt (MPI-MR). Janelia Fluor (JF) HaloTag substrates were kindly provided by L. D. Lavis (Janelia Research Campus, Ashburn, Virginia). The chemical structures of these substrates are listed in Supplementary Fig. 14. All TECAN microplate reader assays were performed using a Spark 20 M instrument (Tecan Group Ltd.). FP assays were performed with either a 384-well microplate (black, non-binding, flat-bottom, low volume, Greiner Bio-One) or a 96-well microplate (flat-bottom, black, Greiner Bio-One). Unless otherwise stated, cpHaloΔ proteins refer to His$_{10}$-tagged proteins after IMAC purification (see Protein purification from *E. coli*).

### Solid phase peptide synthesis
All reagents were purchased from commercial suppliers (BLDPharm, Sigma-Aldrich, Carl Roth GmbH+ Co.KG, Merck KGaA, VWR International). All solvents used for peptide synthesis were anhydrous. The peptide synthesis was carried out on a 0.1 mmol scale per batch. The commercially available Fmoc-Thr(tBu)-Wang resin (Novabiochem, 0.6 mmol/g loading) was used directly.

The automated solid phase peptide synthesis was performed on a CEM Liberty Blue Automated Microwave Peptide Synthesizer, using 0.5 M DIC in DMF as the activator, 0.5 M Oxyma + 0.05 M DIPEA in DMF as the activator base, 0.1 M Oxyma + 20 vol. % of pyridine in DMF as the deprotection solution. The amide coupling was performed under a general condition as: 5.0 equiv. of amino acid solution, 5.0 equiv. of activator and 5.0 equiv. of activator base at 50 °C for 10 min.

Biotin-PEG4-OH was used for N-terminal biotinylation, tetramethylrhodamine 6 carboxylic acid (TMR-6COOH), and Fmoc-PEG2-OH were used for TMR tagging. They were installed by manual SPPS under the following condition: 5.0 equiv. of tagging reagent, 5.0 equiv. of activator and 5.0 equiv. of activator base in DMF/DMSO (1:1, v/v, 3 ml) at RT for 2 h.

The matured peptide was liberated by the treatment with a cleavage cocktail of trifluoroacetic acid/phenol/ water/thioanisole/triisopropyl silane/1,2-ethanedithiol (82:4:4:4:4:2, v/v/v, 5 ml) at room temperature for 2 h. The resulting cleavage solution was collected, and the crude peptide was precipitated by the addition of ice-cold diethyl ether (40 mL). The precipitate was isolated by centrifugation, washed twice with ice-cold diethyl ether (40 mL), and subsequently dried to yield the crude peptide.

Purification of the crude peptide was performed by reverse-phase high-performance liquid chromatography (RP-HPLC) using an Agilent 1260 Infinity II preparative HPLC system equipped with a Shimadzu C18 preparative column (30 × 250 mm, 5 μm pore size). Chromatographic separation was conducted at a flow rate of 40 mL/min using a solvent gradient of 5–40% acetonitrile (MeCN) in water, containing 0.1% (v/v) TFA, over a duration of 60 min. The purity of collected fractions was assessed by HPLC-MS (Supplementary Figs. 15–18), and pure fractions were pooled and lyophilized. The purified peptide was stored at −20 °C until further use.

### Molecular cloning of plasmids for bacterial and mammalian expression
pET-51b(+) vectors (Novagen) was used for cloning constructs for cpHaloΔ protein purification from *Escherichia coli* (*E. coli*), including a N-terminal His$_{10}$-tag for affinity purification, followed by a TEV protease recognition sequence (ENLYFQ|G), without any C-terminal tag. For protein expression in mammalian cells, pcDNA5/FRT or pcDNA5/FRT/TO vectors (Thermo Fisher Scientific) were used for transient transfection or stable cell line generation via Flp-IN TREx system (Thermo Fisher Scientific). pAAVS1-P-MCS (Addgene, #80488) was used for cloning donor template for AAVS1 safe harbor integration[64]. Molecular cloning was performed using Gibson assembly (GA)[65] method or Q5 site-directed mutagenesis (SDM) (NEB) according to manufacturer's protocol. Primers were purchased from Sigma-Aldrich or from IDT (Integrated DNA Technologies). After PCR amplification using KOD-hot-start DNA polymerase master mix (Sigma-Aldrich), DpnI digestion was performed (15 min at 37 °C) to remove template DNA, followed by DNA purification. Home-made competent *E. coli* 10 G or NEB 5-alpha *E. coli* (NEB) cells were used for the transformation of plasmids generated by GA or SDM, respectively. All the constructs were verified by Sanger sequencing (Eurofins).

### Protein purification from *E. coli*
pET-51b(+) plasmids encoding the proteins of interest were transformed into *E. coli* BL21(DE3)-pLysS (Novagen). After overnight preculture at 37 °C, production cultures (0.2–1 L LB-ampicillin medium) were inoculated with a 1:200 dilution ([amp.] = 100 ug/mL). When OD$_{600}$ reached 0.6–0.8, isopropyl $\beta$-D-1-thiogalactopyranoside (IPTG, 0.5 mM) was added, and the temperature was reduced to 16 °C. After 16 h of incubation, cells were collected by centrifugation (4000 RCF, 15 min, 4 °C) and lysed in 30 mL His extract buffer (supplemented with 1 mM PMSF and 0.25 mg/mL lysozyme) by sonication (50% duty, 70% power, 7 min) on ice. After a two-step centrifugation (15,000 RCF, 15 min, 4 °C and 70,000 RCF, 15 min, 4 °C), the protein was purified by immobilized metal affinity chromatography (IMAC) on a gravity flow column or on an ÄktaPure FPLC instrument (Cytiva) with a HisTrap FF crude column (Cytiva). Recipes for His washing and His elution buffer are provided in Supplementary Table 5. Amicon Ultra-15 Centrifugal Filter Units (10 kDa MWCO) were used for buffer exchange to activity buffer (recipes provided in Supplementary Table 5) and concentrating proteins. Protein molecular weight and purity were validated by SDS-PAGE and high-resolution mass spectrometry (HRMS).

### SDS-PAGE and in-gel fluorescence assay
Samples for SDS-PAGE were prepared using sample buffer (recipe provided in Supplementary Table 5) and denatured by heating at 95 °C for 10 min. Precast gels (4–20% Mini-PROTEAN® TGX Stain-Free™ Protein Gels, Bio-Rad) were run at 200 V for 20 min. Imaging was performed on a ChemiDoc™ Imager (Bio-Rad) post UV activation allowed by the stain-free Biorad technology. Fluorescently labeled proteins (SiR and CPY) were imaged in Cy5 channel (red epi illumination: 625–650 nm excitation and 695/55 emission filter) using the same imager.

## Preparation of fluorescently labeled proteins

For fluorescence profile evaluation, $His_{10}$-tagged cpHaloΔ variants (50 μM) were labeled in presence of Hpep1 (100 μM), CA-CPY or CA-SiR (75 μM) in activity buffer for 60 h at 4 °C. No Hpep1 was added for preparing labeled HaloTag.

For affinity measurements, $His_{10}$-tagged cpHaloΔ3 (50 μM) was labeled using Hpep1 (10 μM) and CA-SiR or CA-TMR (100 μM) for 85 h at 4 °C. After incubation, Hpep1 was removed by IMAC purification on a gravity flow column and buffer was exchanged to activity buffer using Amicon Ultra-15 Centrifugal Filter Unit. The protein labeling was verified by SDS-PAGE, comparing Coomassie blue staining and in-gel fluorescence (Supplementary Fig. 19). Protein labeling and Hpep1 removal were further verified by HRMS as previously explained.

## Protein concentrations

Protein concentrations were determined after centrifugation (14,000 rpm = 18,407 RCF, 4 °C, 10 min) by measuring absorption at 280 nm with a NanoDrop 2000c spectrometer (Thermo Fisher Scientific). For proteins labeled with CA-fluorophores, protein concentrations were determined by measuring absorption at 280 nm and the maximal absorption of the corresponding fluorophore. The concentrations of the labeled proteins were corrected using the correction factor at $A_{280}$ of the different fluorophores using Eq. (1):

$$Protein\ concentration\ (M) = \frac{A_{280} - (A_{max} \times CF)}{\varepsilon_{protein}} \times dilution\ factor$$

(1)

where:

$A_{max}$ = absorbance at 555 nm for TMR- and 652 nm for SiR-cpHaloΔ3

CF = correction factor (0.34 for TMR and 0.147 for SiR)

$\varepsilon_{protein}$ = protein extinction coefficient

## Fluorescent substrate concentrations

Fluorescent substrates were dissolved in anhydrous DMSO as stock solutions, stored at −20 °C and subsequently diluted into specific buffers or media freshly, ensuring the final concentration of DMSO remained below 1% (v/v). The concentrations were determined by measuring absorbance at the corresponding maximal absorption wavelength for each fluorescent substrate in PBS pH 7.4 (Gibco) or PBS with 0.1% (w/v) SDS, by measuring their UV-Vis spectra on a NanoDrop 2000c spectrometer (Thermo Fisher Scientific). Concentrations were calculated using the Lambert−Beer's Law using the extinction coefficients characterized in-house or as previously reported[66,67] (Supplementary Table 7).

## Thermostability analysis

The melting temperature (mT, half-denaturation temperature) of the cpHaloΔ variants were determined by following fluorescence intensity ratio change at 350 and 330 nm from 20 °C to 95 °C (1 °C/min) using a nanoscale differential scanning fluorimeter (nanoDSF) Prometheus NT 48 device (NanoTemper Technologies). The inflection point of the ratio curve (maximum of the first derivative) corresponds to the melting temperature of the protein (mean ± s.d., N = 2).

## Yeast display library preparation

Primers containing degenerated codon NNKs were purchased from Sigma-Aldrich. The pJYDNG vector[68] (Addgene, #162452) was used to create cpHaloΔ plasmid (pJYDNg-appS4-NcoI-cpHaloΔ-NotI-Aga2p-HA-myc-eUnaG2) containing cpHaloΔ flanked by two restriction enzyme (RE) digestion sites, NcoI and NotI. This plasmid was used as template for library preparation using self-ligation of the whole-plasmid PCR product[69] as described previously. The recipe for each reaction can be found in Supplementary Tables 8–11.

PCR amplification was performed using the KOD-hot-start DNA polymerase master mix (Sigma-Aldrich) with a reaction scale of 800 and 600 μL for N- and C-terminal extension libraries, respectively. DpnI digestion was performed for the resulting PCR products, followed by DNA purification with two washing steps with PE buffer. The purified DNA was digested with NcoI or NotI (NEB) for N- or C-terminal extension library, respectively. DNA purification was conducted before ligation using T4 DNA ligase (NEB) following standard procedure with an incubation at room temperature (RT) for 1.5 h. After ligation, the ligase was digested by protease K (NEB) at 37 °C for 30 min, followed by heat-inactivation at 65 °C for 15 min. DNA was then concentrated by ethanol precipitation by supplementing DNA with 10 μL 3 M sodium acetate, 100 μL isopropanol, and glycogen (ThermoFisher) at a final concentration of 0.5 μg/μL. After more than 20 h of incubation at −20 °C, the mixture was centrifuged (10,000 rpm = 9391 RCF, 15 min) and the pellet was washed with 1 mL ice-cold 70% ethanol. After another centrifugation (10,000 rpm = 9391 RCF, 10 min), the pellet was air-dry for further use. The resuspended DNA was transformed into home-made *E. coli* 10 G by electroporation. Cells were recovered in LB medium at 37 °C for 1 h at 220 rpm. 5% of the cells were used for library size determination through serial dilutions and plating on kanamycin (50 μg/mL) selection agar plate. The rest of the cells were inoculated into 50 mL LB-kanamycin medium at 37 °C and 220 rpm for overnight culture and plasmid extraction.

Electro-competent *Saccharomyces cerevisiae* strain EBY100 (ATCC) cells were prepared according to a literature procedure[70] with few modifications. A single colony was used to inoculate a 50 mL yeast extract peptone dextrose (YPD) pre-culture, grown at 30 °C and 250 rpm. A 100 mL YPD culture was inoculated with an initial $OD_{600}$ of 0.1 and grown until $OD_{600}$ reached 1.3–1.5. At this moment, Tris-DTT (800 μL) and Tris-LiAc (2 mL) were supplemented into the culture and incubated for 15 min at 30 °C, 250 rpm. The cells were then collected by centrifugation (2500 RCF, 4 °C, 3 min), washed with ice-cold electroporation buffer, centrifuged (2500 RCF, 4 °C, 3 min) and resuspended in 300 μL electroporation buffer. An aliquot of 50 μL cells was used for a single electroporation (1–3 μg DNA). Electroporation was performed on a Gene Pulser Xcell System (0.54 kV, 25 μF, infinite resistance with an exponential decay, Bio-Rad) with a pre-chilled electroporation cuvette (GenePulser® Cuvette, 0.2 cm gap, Bio-Rad). Yeast cells were recovered in YPDS medium at 30 °C and 250 rpm for one hour, harvested by centrifugation (2500 RCF, 5 min), and resuspended in 1 mL SDCAA medium, from which the 5% of the cells were used for plating serial dilutions to determine library size. The rest of the cells were then transferred into 100 mL SDCAA culture and propagated at 30 °C and 250 rpm.

## Expression and labeling cpHaloΔ at yeast surface

A saturated yeast culture (0.5 mL) was inoculated into SGCAA medium (4.5 mL) to induce protein expression at 30 °C and 250 rpm for 20 h. $10^7$ cells ($OD_{600}$ = 1 corresponds to $10^7$ cells/mL[70]) from the induced culture were harvested by centrifugation (14,000 RCF, 1 min). Yeast cells were resuspended in 50 μL PBS and stained with HaloTag ligands and additional reagents allowing for later sorting. Cells were washed by centrifugation (14,000 RCF, 1 min) and resuspension in PBS (125 μL). Yeast cells were incubated with bilirubin (100 nM) on ice for 15 min to label eUnaG2, used as expression marker. Prior to fluorescence-activated cell sorting (FACS), yeasts were resuspended in 2 mL PBS and filtered into a 5 mL round bottom polystyrene test tubes through a cell strainer cap (Falcon, #352235).

In the first round of screening, yeast libraries, or yeast expressing parental cpHaloΔ as control, were labeled with 1 μM CA-TMR in the presence and absence of 2 μM Hpep9 for 30 min at RT on a rotating wheel and the reaction was quenched with 5 μM HaloTag protein for 10 min. In the following rounds, labeling was performed with a two-step labeling procedure. In the first step, CA-CPY (5 μM) was added in

the absence of Hpep9 for 30 min and the reaction was subsequently quenched with HaloTag protein (10 μM) for 10 min. After washing twice with PBS, cells were incubated with CA-TMR (1 μM) in the presence or absence of Hpep9 (detailed conditions are provided in Supplementary Tables 13 and 14), and the reaction was further quenched by addition of HaloTag (10 μM) for 10 min. Finally, the cells were washed once prior to bilirubin labeling.

## Yeast library screening by FACS

Labeled yeast cells were sorted on a BD FACSMelody Cell Sorter (BD Biosciences) with a neutral density filter 1.5 (FSC ND filter) and a 100 μm sorting nozzle. The corresponding lasers and filters for each fluorophore can be found in Supplementary Table 11. The gating strategy and the detailed information for each screening round can be found in Supplementary Fig. 20 and Tables 12 and 13. Counter gating was used for selecting cells with highest TMR signal in the low CPY population. For N-terminal extension library, a negative sort was performed at the 5th round on cells labeled with CPY and TMR both in the absence of Hpep9. The cells showing low signal in both TMR and CPY channel were sorted. Afterwards, the sorted cells were grown in 5 mL SDCAA medium containing penicillin-streptomycin (50 U/mL, Gibco) for two days. Cells were passaged once by inoculating 0.5 mL culture into 4.5 mL SDCAA medium before induction of protein expression. At each screening round, plasmids from the sorted population were extracted from 1 mL saturated yeast culture using Zymoprep Yeast Plasmid Miniprep II kit (Zymo Research) following manufacturer's procedure. The resulting plasmids were subsequently amplified by retransformation into *E. coli* 10 G for miniprep preparation.

## Next-generation sequencing (NGS)

The region containing randomization residues (150 bp) was amplified by PCR reaction at 50 μL scale with 0.5 μL template (2 ng/μL) and the corresponding primers (Supplementary Table 15), followed by DNA purification. Barcode-containing primers (NGS adaptor ligation oligos) were purchased from Eurofins Genomics. PCR products from different screening rounds and from different libraries were pooled in equivalent quantities and submitted for NGS sequencing (Illumina technology, Eurofins Genomics), targeting 5 million read pairs (2 × 150 bp). The use of unique barcodes for each screening round and library enabled subsequent demultiplexing of the sequencing data.

NGS results were analyzed using an in-house pipeline with the following procedure. The quality of sequencing was first checked using FastQC tool[71]. DNA reads were then grouped according to their barcodes using (je demultiplex[72]) and forwards and reverse reads were combined as a single sequence for each barcode. To trim off the barcodes, annealing region, and the adapter sequences, (Cutadapt[73]) was used. The trimmed reads were subsequently aligned to the reference template using bowtie2 short read aligner[74] and samtools[75] was used to convert the resulting "sam" files into "bam" files. A custom-made R script was then used to calculate amino acid frequencies at each randomization position by dividing the number of reads containing the specific residue at that specific position by the total number of reads.

## Labeling kinetics measurement of split-HaloTag

The labeling kinetics of split-HaloTag with CA-TMR was measured by recording fluorescence polarization (FP) over time using a TECAN microplate reader. The reactions were prepared in a 384-well plate with a final volume of 40 μL. 20 μL Hpep at varying concentrations (twofold dilution starting at 1 μM final) and 10 μL cpHaloΔ protein (2.5 or 20 nM final) was mixed in the 384-well plate, followed by a brief centrifugation (500 RCF, 1 min). Immediately before the measurement, CA-TMR (0.5 or 4 nM final) was added using a 384-channel pipetting robot (VIAFLO, Integra Biosciences GmbH). FP was recorded (excitation filter: 535/25 and emission filter: 595/35) at 37 °C for a period of 10 h with a humidity cassette (Tecan Group Ltd.) to reduce

the effect from evaporation for long-term measurements. The raw data was corrected for the time-delay that occurred between the addition of CA-TMR and recording of the first point. The raw data of replicates were first averaged and the data was fitted with Eq. (2) for calculating the estimated apparent second-order rate constant $k_{app}$.

$$FP(t) = FP_{bound} + \frac{FP_{free} - FP_{bound}}{[A]_0} \cdot \frac{[A]_0([A]_0 - [B]_0)e^{([A]_0-[B]_0)k_{app}\cdot t}}{[A]_0 \cdot e^{([A]_0-[B]_0)k_{app}\cdot t} - [B]_0} \quad (2)$$

where:

t = time
FP(t) = FP at time t
$FP_{free}$ = FP of the free CA-TMR
$FP_{bound}$ = FP of the bound TMR fluorophore
$[A]_0$ = CA-TMR concentration at t = 0
$[B]_0$ = protein concentration at t = 0
$k_{app}$ = apparent second-order rate constant

In the conditions where the reactions did not plateau, a linear model was used for data fitting. The initial slopes of the fitted curves ($s_{t=0}$) were determined using a previously reported a R script[36]. Monte Carlo simulations were employed to evaluate the uncertainties and determine the confidence intervals for the fitted values (N = 1000, 5% worst fits were discarded).

## Half-maximal complementing concentrations (EC50) determination

EC50 of peptides in the split-HaloTag context was determined by performing labeling kinetics at different peptide concentrations. EC50 was defined as the Hpep concentration required to reach 50% of the maximal labeling speed for a given split-HaloTag pair. It was used as a proxy for the affinity between the split-HaloTag fragments. For EC50 determination, the initial slopes (mean and standard deviation, s.d.) of each labeling condition were plotted against a logarithm of the corresponding Hpep concentration. The sigmoidal model was fitted to the data using R script (for data shown in Supplementary Fig. 3a, b) with Eq. (3) and the confidence intervals of the fitted values were estimated with the Monte Carlo approach[76].

$$k_{app}([Hpep]) = \frac{k_{max}}{1 + 10^{(\log_{10}(EC_{50}) - \log_{10}([Hpep]))}} \quad (3)$$

where:

$k_{app}$ = apparent second-order rate constant
$k_{max}$ = $k_{app}$ at saturating Hpep concentration (maximal $k_{app}$)
$EC_{50}$ = half-maximal effective concentration
[Hpep] = the concentration of Hpep

Alternatively, EC50 was calculated using GraphPad Prism (version 9.2.0) with a sigmoidal dose-response curve (fixed Hill slope = 1.0). The data was fitted with Eq. (4):

$$Y = Bottom + X \cdot \frac{(Top - Bottom)}{(EC_{50} + X)} \quad (4)$$

where:

Y = initial slope
X = corresponding concentration of Hpep

## Background labeling activity of cpHaloΔ in vitro

The background labeling kinetics of cpHaloΔ variants with CA-TMR in the absence of Hpep were recorded using a TECAN microplate reader. Twofold serial dilutions of cpHaloΔ ranging from 0.2 to 200 μM (final, 20 μL) was prepared in a 384-well plate and a brief centrifugation (500 RCF, 1 min) was performed. Afterwards, 20 μL of CA-TMR (50 nM final) was added using the pipetting robot immediately prior FP measurement for 10 h at 37 °C with a humidity cassette. The raw data was fitted

with a nonlinear regression with a pseudo-first order association equation for calculating the half-time of the labeling using GraphPad Prism (version 9.2.0).

## Fluorescence intensity scan

To determine the influence of Hpep variants on the apparent fluorescence intensity of the split-HaloTag/dye complex, a reaction was prepared in activity buffer (supplemented with 0.1 mg/mL BSA) with the labeled cpHaloΔ (0.2 μM) and the corresponding Hpep variant (1 μM) and incubated at 37 °C for 5 h in a 96-well plate. A fluorescence intensity (FI) scan was performed at 37 °C, covering 600–700 nm (5 nm step, 10 nm bandwidth) with the excitation at 560 nm (20 nm bandwidth) on a TECAN plate reader.

## Affinity ($K_d$) determination using fluorescence polarization assay

To use FP as a readout to determine the affinity between cpHaloΔ variants (cpHaloΔ2, cpHaloΔ3, cpHaloΔ3-SiR) and Hpep, we chemically synthesized TMR-Hpep conjugates (TMR-Hpep9, TMR-Hpep11). A TMR-Hpep (1 nM) was titrated against a cpHaloΔ variant (fivefold serial dilutions) in activity buffer in a 384-well microplate. FP was measured using a TECAN plate reader and the following settings: excitation filter (535/25), emission filter (595/35), 30 flashes, 40 μs integration time. Technical triplicates were performed for each condition. The FP values were plotted against the logarithm of the corresponding cpHaloΔ protein concentration and the nonlinear regression model (three parameters, Hill slope = 1.0) was used for data fitting to get $EC_{50}$ values, considered to be $K_d$ values.

## Binding kinetics analysis by bio-layer interferometry

Bio-layer interferometry (BLI) analysis was performed in a 96-well plate using an Octet® R4 system (Sartorius) with an acquisition rate for standard kinetics of 5.0 Hz at 25 °C. Chemically synthesized biotin-Hpep conjugates (biotin-Hpep9, biotin-Hpep11) were immobilized onto streptavidin (SA) biosensor surface in activity buffer, supplemented with 0.05% BSA, which was the buffer used for all measurement steps, including: (i) biosensor hydration in buffer for 10 min; (ii) baseline recording for 60 s; (iii) biotin-Hpep (50 nM) immobilization for 300 s; (iv) baseline recording for 60 s; (v) association step with varying concentration of cpHaloΔ3 proteins (300–2 nM) for 60 or 300 s; (vi) dissociation step in buffer for 300 s. The raw data was analyzed using Octet® Analysis Studio (Sartorius). For data correction, response curves were baseline-aligned by subtracting the average signal of the baseline step and Savitzky–Golay filtering was used to remove high-frequency noise from the data. Data from the measurements of different cpHaloΔ protein concentrations were grouped and a 1:1 binding model was used for a global data fitting (kinetic fit) to generate the values of $K_d$, $k_{on}$, and $k_{off}$, from which $K_d$ could be extrapolated.

## Molecular dynamics simulation

Molecular dynamics (MD) simulations were performed for 200 ns under constant pressure (NPT) to examine the interactions between Hpep11 and cpHaloΔ variants. The protein–peptide complexes were generated using AlphaFold3 (AF3)[39], developed by Google DeepMind in collaboration with Isomorphic Labs with FASTA-formatted input sequences. Predictions were performed using the AF3 web platform with default settings unless otherwise specified. The top-ranked models of Hpep11/cpHaloΔ were further processed using the Protein Preparation Wizard (Schrödinger Inc., New York) with default settings to optimize the structure for molecular dynamics simulations. An orthorhombic periodic box was generated with a 10 Å buffer distance via the Desmond System Builder module in Maestro (Schrödinger Inc., New York). Sodium and chloride ions were introduced into the solvent phase to neutralize the system and achieve an experimental salt

concentration of 0.15 M NaCl. The simulation system, comprising approximately 30,000 atoms, was constructed using the OPLS4 force field. MD simulations were performed using the default algorithms and protocols of the Desmond Molecular Dynamics module (Schrödinger Inc., New York). The system was equilibrated before executing a 200 ns production simulation at 300 K and 1.01325 bar. Data analysis was conducted using the Simulation Interaction Diagram tool within Maestro.

## General information for mammalian cell culture

Mammalian U-2 OS Flp In™ T-REx™ cells[77] (referred to as U2OS cells) and Flp-In™ T-REx™ 293 cells (hereafter referred to as 293FT cells, Invitrogen™) were cultured in growth medium, filtered with Nalgene™ Rapid-Flow™ Sterile Filter (0.22 μm, 500 mL, Thermo Fisher Scientific). Cells were passaged every 2–4 days or at confluency and grown at 37 °C with 5% $CO_2$ by washing with PBS (Gibco), digestion with TrypLE™ Express (Gibco) and resuspension in growth medium. *Mycoplasma* contamination tests were performed regularly on all cell lines. For long-term storage at −80 °C, U2OS cells and 293FT cells were stored in 95% growth medium with 5% DMSO or 90% FBS with 10% DMSO, respectively. Cell counting was conducted using Countess 3 automated cell counter (Invitrogen) with a Countess 3 cell counting chamber slide (Invitrogen). Plasmids for mammalian cell expression were cloned in pcDNA™5/FRT/TO vectors (Thermo Fisher Scientific), with which the expression of genes of interest (GOIs) were driven by CMV promoter. To induce protein expression, 500 ng/mL doxycycline was added with at least overnight incubation. All the stable cell lines used in this study were sorted using a FACSMelody™ Cell Sorter (BD) and are listed in Supplementary Table 16.

## Transient transfection in mammalian cells

Mammalian cells were seeded onto specific plates depending on the application with 8–20 k cells per well with addition of doxycycline in the case of the inducible cell lines. When cells reached a confluency of about 70%, Lipofectamine 3000 Reagent (Thermo Fisher Scientific) was used for transfection. In a tube, 100 ng of plasmid DNA was mixed with 10 μL OptiMEM and supplemented with 0.2 μL of P3000. In another tube, 0.2 μL of Lipofectamine 3000 was diluted in 10 μL of OptiMEM. Next, the contents of the two tubes were combined and incubated at RT for 15 min before adding it to the cells. After 4–8 h of incubation, the media was replaced with fresh growth medium and cultured for 24–48 h prior to imaging or flow cytometry analysis. Plasmids used for transient transfection in each experiment are listed in Supplementary Table 17.

## Cell fixation and staining

Paraformaldehyde (PFA) fixation was used in this study. Cells were washed with pre-warmed PBS and fixed with 4% (w/v) methanol-free PFA in PBS for 15 min at 37 °C. Afterwards, cells were washed with PBS for three times for further analysis.

## Live-cell staining with fluorescent substrates

In this study, the cells were stained with fluorescent substrates (100 nM, unless otherwise stated) that are specific to the self-labeling proteins (SLPs), in growth medium for wash experiments and in imaging medium for no-wash experiments. For wash experiments, cells were washed twice with imaging medium with a 5 min incubation at 37 °C for each washing step.

## Immunostaining with recombinant cpHaloΔ

U2OS cells stably over-expressing either H2B-SNAPf-Hpep11 or TOM20-SNAPf-Hpep11, generated by Flp-IN integration, were seeded on ibidi 8-well glass-bottom imaging dish and the protein expression was induced by the addition of doxycycline for 24 h. After fixation, cells were permeabilized with 0.2% Triton ×-100 in

PBS for 15 min at RT. Afterwards, cells were washed with PBS containing 1% BSA (w/v) and the blocking reagent (5% BSA (w/v) in 0.2% Tween 20) was supplemented. After 30 min incubation at RT, cells were incubated with cpHaloΔ3 (1 µM), a CA-fluorophore (CA-SiR or CA-CPY, 500 nM) and a SNAPf substrate (CP-TMR, 500 nM) at 4 °C for overnight. The next day, cells were washed twice with PBS containing 1% BSA (w/v) before imaging on a Stellaris 5 microscope. U2OS blank cells were also treated with the same procedure to reveal the absence of unspecific binding of the fluorophores or of labeled cpHaloΔ3 protein.

### Stable cell line generation using Flp-IN TREx system

For the stable cell lines generated using Flp-IN™ T-REx™ system (Thermo Fisher Scientific), cells were grown in a T25 culture flask (Sarstedt AG & Co.KG) until ~80% confluency. Lipofectamine 3000 transfection, as described above, was performed by co-transfecting the pCDNA5/FRT plasmid encoding POI (440 ng) and the pOG44 plasmid encoding the Flp-recombinase (3560 ng, Invitrogen). 8 µL of Lipofectamine 3000 was diluted in 200 µL OptiMEM. DNA mixture was diluted in 200 µL OptiMEM and supplemented with 8 µL of P3000 reagent. After overnight incubation with the transfection reagent, the medium was replaced by fresh growth medium. After a recovery period of 8–24 h, hygromycin B (100 µg/mL, Gibco Life Technologies) was added to select positive cells with stable integration of the cassette into the genome for 48–72 h. The cells were then recovered in fresh growth medium and sorted by FACS upon reaching confluency to select a bulk population with a positive signal for EGFP and/or the labeled-SLP. Doxycycline was added one day prior to sorting to induce protein expression.

### Stable cell line generation by AAVS1 safe harbor integration

Stable cpHaloΔ3 cell lines (U2OS and 293FT cell lines) were alternatively generated by stable integration of a cassette encoding the POI into the AAVS1 safe-harbor locus of the human genome[64] using a CRISPR/Cas9 KI approach. For this, the GOI was cloned into the pAAVS1-p-MCS plasmid (Addgene, #80488) by Gibson assembly, as described in the previous section. Lipofectamine 3000 transfection was performed on cells cultured in a T25 flask, as described above, by co-transfecting the cells with 2 µg of the pAAVS1-p-POI plasmid and 2 µg of the pXAT2 plasmid (Addgene, #80494), which encodes Cas9 and the guide RNA targeting the AAVS1 genomic locus. Selection of positive cells with stable integration of the GOI was achieved by supplementing the growth medium with puromycin (1 µg/mL) for 3 days. Bulk sorting for EGFP-positive cells was performed by FACS when the cells reached confluency. The day before sorting, doxycycline was added to induce protein expression.

### Pipeline for generating the endogenous Hpep-tagged cell lines

In U2OS cells, the endogenous Hpep11-tagged cell lines used in this study were generated in two different procedures, depending on the parental cell line employed for introducing the Cas9/gRNA complex (either U2OS blank cells or cpHaloΔ3-T2A-NLS-EGFP stable cell line generated by Flp-In T-REx system). In 293FT cells, genome editing was performed on a single clone of a stable cell line, which was generated by the stable integration of cpHaloΔ3-T2A-NLS-EGFP into the AAVS1 safe-harbor locus. Genome editing was performed using Lipofectamine™ CRISPRMAX™ Cas9 Transfection Reagent (Invitrogen) for the delivery of a complex of donor DNA and ribonucleoprotein (RNP), which consists of crRNA to target a specific genomic locus[12,29,78], a Cas9 binding scaffold (tracrRNA), and Cas9 protein. The crRNA (Alt-R® CRISPR-Cas9) and ssDNA donors encoding Hpep (Ultramer™ DNA oligo) were purchased from IDT (Supplementary Table 3). crRNAs were resuspended in nuclease-free duplex buffer (IDT) at 100 µM and stored at −20 °C and the concentration of donor DNA was adjusted to 100 µM.

On the day of transfection, duplex gRNA was prepared by heating the complex of crRNA/tracrRNA-Atto647 (1 µM) at 95 °C for 5 min, followed by cooling to RT and Cas9-RFP protein was diluted in Opti-MEM (1 µM). Cells were seeded on a 6-well plate (Nunc™, Thermo Fisher Scientific) to reach ~80% confluency on the day of transfection. In a tube, 24 µL of duplex gRNA, 24 µL of Cas9-RFP, 9.6 µL of Cas9 Plus Reagent were added into 400 µL of OptiMEM, followed by supplementation of 3 µL of donor ssDNA (prepared in sterile water). In another tube, 19.2 µL of Lipofectamine CRISPRMAX Reagent was added into 400 µL of OptiMEM. The contents from the two tubes were mixed and incubated at RT for 15 min. The cell culture medium was replaced with fresh medium containing 1 µM of HDR Enhancer V2 (IDT), and the transfection mixture was then added to cells. After overnight incubation, cell culture medium was replaced with fresh medium. FACS sorting was performed 14–24 h post-transfection to enrich for cells transfected with both Cas9-RFP and tracrRNA-Atto647. The sorted cells were then cultivated for a period of 5–7 days. Following this, a second round of FACS was performed after one-hour labeling with CA-SiR (100 nM) to enrich the cell population for both Hpep and cpHaloΔ3 integration. The gating strategy is detailed in Supplementary Fig. 21.

### Generation of the endogenous HaloTag-tagged cell lines

The endogenous HaloTag-tagged cell lines were generated using the same procedure as described above, with the exception of the HDR template preparation. This was accomplished either by purchasing from IDT as a gBlock or by PCR amplification using primers containing homology arms. After PCR amplification, DNA fragments with correct size were purified from an agarose gel using the QIAquick Gel Extraction Kit (QIAGEN). Cas9/gRNA and the donor DNA was delivered into blank U2OS or 293FT cells using Lipofectamine™ CRISPRMAX™ Cas9 Transfection Reagent (Invitrogen), according to the previously detailed procedure whereby 1200 ng donor DNA was used. Two rounds of cell sorting using FACS were performed: the first round enriched for transfected cells, and 5-7 days later, the second round selected for cells with HaloTag integration by gating SiR-positive cells after one-hour labeling with CA-SiR (100 nM).

### Flow cytometry analysis

U2OS stable cell lines were seeded onto a 96-well clear flat-bottom plate (Fisher Scientific GmbH) with 100 µL of growth medium containing doxycycline to gain ~90% confluency on the day of analysis. After labeling with HaloTag ligand (100 nM), cells were washed once with PBS and digested with 50 µL of TrypLE™ (Gibco). After 10 min of incubation, 150 µL FACS buffer was added and the samples were then transferred onto a U-bottom 96-well plate (Falcon). The samples were measured via LSRFortessa™ ×-20 Cell Analyzer (BD Biosciences) with the with a high throughput sampler (HTS) for 96-well plates (flow rate: 3 µL/s; 3 mixing steps; wash volume: 400 µL). Photomultiplier tubes (PMT) settings were optimized using the positive and negative controls. Optical settings for each fluorophore are listed in Supplementary Table 12. Data analysis was performed using FlowJo™ software (BD Biosciences). The labeling procedure for investigating cpHaloΔ labeling kinetics in cells (Supplementary Fig. 4) was performed with CA-SiR or CA-CPY (100 nM final) for eight different time periods ranging from 5 min to 20 h. At the end of incubation period, recombinant HaloTag protein (1 µM final) was added and incubated to quench the labeling reaction. To label TOM20 tagging cell lines (Fig. 3b and d and Supplementary Fig. 12b), cells were labeled with CA-SiR (100 nM final) for one hour.

### Confocal fluorescence microscopy

Confocal microscopy was conducted using a Leica Stellaris 5 equipped with a supercontinuum white light laser (470–670 nm), hybrid

photodetectors for single-molecule detection (HyD SMD) and an environmental chamber at 37 °C with 5% $CO_2$. The laser power output was adjusted to 85% of the maximum and calibrated regularly. Cells were seeded into imaging plates with glass bottoms (μ-Slide 8 Well, μ-Slide 18 Well, or μ-Plate 96 Well, ibidi GmbH). Unless otherwise stated, cells were stained with 100 nM fluorescent substrates (CA-fluophores, CP-fluorophores) for one hour and the images were taken with a 40× (HC PL APO CS2 40×/1.10) water objective, with a scan speed of 400 Hz, line average of 2 and 12-bit pixel depth (specific imaging acquisition parameters are listed in Supplementary Table 17).

### Expansion microscopy
U2OS stable cell lines were seeded onto CultureWell™ Chambered Coverglass plate (Invitrogen) and induced with 1000 ng/mL doxycycline in order to reach ~90% confluency on the day of labeling. The cells were labeled with HaloTag substrates (CA-MaP555, CA-TMR, or CA-CPY, 100 nM) for 2 h, then washed twice with imaging medium. ExM sample preparation was performed following a literature protocol with few modifications[79]. Cells were fixed with 4% (w/v) formaldehyde (FA) in PBS for 15 min at 37 °C, washed twice with PBS, then incubated in 0.7% (w/v) FA and 1% acrylamide (AA) in PBS for 5 h at 37 °C. After PBS washes, gelation was performed with U-ExM monomer solution supplemented with 0.5% (w/v) ammonium persulfate (APS) and 0.5% (w/v) tetramethylethylenediamine (TEMED). Cells were incubated on ice for 5 min, then at 37 °C for 1 h. The gel samples were transferred into denaturation buffer at RT for 15 min with gentle shaking. The samples were then moved into pre-heated fresh denaturation buffer at 95 °C for 15 min. Following denaturation, samples were expanded in deionized water, with water changed every 20 min until the expansion was completed. The expanded gels were re-stained with Hoechst 33342 (5 ug/mL) to facilitate locating the cells, mounted with 2% (w/v) low-melt point agarose, and stored until imaging.

### Live-cell stimulated emission depletion (STED) nanoscopy
U2OS cell lines were seeded in 8-well glass-bottom imaging plate (Ibidi) and prepared as previously described. Live-cell STED nanoscopy was conducted with an Abberior STED Expert Line 595/775/RESOLFT QUAD scanning microscope equipped with a UPlanSApo 100×/1.4 oil immersion objective lens (Abberior Instruments) without temperature and $CO_2$ control. The 640 nm excitation lines and the 775 nm STED lines were used for SiR, collecting emission at 650–757 nm on avalanche photodiodes (APDs). Image acquisition parameters are described in detail in Supplementary Table 18. The laser powers were regularly monitored at the back focal aperture and remained constant during the duration of experiments. Laser powers are given in % and absolute laser powers in μW (640 nm excitation) and mW (775 nm STED) are given in Supplementary Tables 19 and 21. For data representation, pixel intensities were scaled between dark and bright colors using the "Red Hot" or "gray" lookup tables (Fiji version 1.54f) as indicated by a reference bar. For image representation, "Gaussian blur" filter (Fiji version 1.54 f) with radius 1 was used.

The performance of split-HaloTag in STED and confocal microscopy was assessed by calculating the Full Width at Half Maximum (FWHM) of single microtubules revealed by Hpep11-*TUBB4B* and HaloTag-*TUBB4B*. Fluorescence intensity profiles perpendicular to microtubules were extracted using ImageJ2 (version 2.3.0). The data were fitted with the GraphPad Prism software (version 10.3.0) and the Gaussian equation (Eqs. (5 and 6)).

$$y = y_0 + \frac{A}{\omega\sqrt{\frac{\pi}{2}}} e^{-2\frac{(x-x_c)^2}{\omega^2}} \tag{5}$$

y: fluorescence intensity, $y_0$: offset, x: x-coordinate, $x_c$: center, A: area, ω: width.

$$FWHM = \omega\sqrt{2\ln(2)} \tag{6}$$

FWHM: Full Width at Half Maximum, ω: Gauss width.

### Fluorescence lifetime imaging
Live-cell fluorescence lifetime imaging was performed on a Leica SP8 FALCON microscope equipped with a Leica TCS SP8 X scanning, SuperK white light laser at pulse frequency of 80 or 40 MHz. To detect SiR labeling signal, excitation was performed at 640 nm and emission was collected within the range of 650–720 nm. Ibidi 8-well glass-bottom imaging plates were used for seeding cells. Mean fluorescence lifetimes calculation was done in the LAS X software (Leica Microsystems) by fitting mono-exponential decay model to the decay.

Fluorescence lifetime of SiR-labeled cpHaloΔ3 was determined in stable U2OS cell line expressing cpHaloΔ3-T2A-NLS-EGFP, with an overnight incubation of CA-SiR (100 nM). Mean fluorescence lifetimes (intensity weighted) were calculated from five cells, with over four regions of interest (ROIs) selected for each cell. To determine the fluorescence lifetime of each Hpep variant, transient transfected U2OS cells expressing H2B-SNAPf-Hpep were stained with CA-SiR (100 nM) for one hour before imaging. 1000 photons per pixel were collected. Mean fluorescence lifetime values (intensity weighted) were calculated from 5 to 10 cells, with over 4 ROIs selected for each cell.

The separation of the background signal from the unbound SiR-labeled cpHaloΔ3 was conducted by positioning the cluster-circles on the phasor plot centering at 2.2–2.5 ns and another cluster-circle at the position where the SiR-cpHaloΔ/Hpep11 complex was locate using Leica Microsystems. To separate signals from Hpep11 and one of the other Hpep, the cluster-circles were centered at the positions of the pure species on phasor plot. The detailed image acquisition parameters can be found in Supplementary Table 18.

### Statistics and reproducibility
All the in vitro and flow cytometry measurements included in this study were at least performed in technical duplicates. All the fluorescence imaging experiments were repeated for at least two times in separate days to ensure similar results. Sample size of each experiment is indicated in the corresponding figure legend.

### Reporting summary
Further information on research design is available in the Nature Portfolio Reporting Summary linked to this article.

## Data availability
All data generated or analyzed during this study are included in this published article (and its Supplementary Information files). The raw datasets for all figures are provided in the Source Data files. All the cell lines generated in this study are listed in the Supplementary Information file. Plasmids described in this study are available from the corresponding author upon request. The Supplementary Videos of the molecular dynamic simulations are available at Zenodo (DOI: 10.5281/zenodo.18754293). No statistical methods were used to predetermine sample sizes. Source data are provided with this paper.

## Code availability
The R analysis pipeline used for FP kinetics curve fitting (and the original, unedited FP kinetic data as TECAN export files) are publicly available at https://github.com/yin-hsi/splitHaloTag_FP_kinetics_fitting and at Zenodo (DOI: 10.5281/zenodo.18750094).

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

## Acknowledgements

This work was supported by the Max Planck Society and École Polytechnique Fédérale de Lausanne (EPFL) and the Deutsche Forschungsgemeinschaft (DFG, German Research Foundation) SFB TRR 186. Y.-H.L. was supported by EPFL Doctoral Program in Biotechnology and Bioengineering (EDBB). J.K., J.W., and S.K. were supported by Heidelberg Biosciences International Graduate School (HBIGS). D.S. was supported by a Humboldt Research Fellowship and an interinstitutional postdoctoral fellowship of The Health + Life Science Alliance Heidelberg Mannheim. J.W. and K.H. were supported by the Max Planck School Matter to Life. We thank L. D. Lavis for providing Janelia Fluor HaloTag substrates and A. Bergner, B. Réssy, D. Schmidt for providing reagents. We thank S. Fabritz, T. Rudi, and J. Kling from the mass spectrometry facility of MPIMR for their support. We thank M. Tarnawski (protein expression and characterization facility, MPIMR) for performing nanoDSF measurements and Elisa D'Este (optical nanoscopy facility, MPIMR) for support.

## Author contributions

Y-H.L., K.J., and J.H. conceived and planed the project. Y-H.L. carried out the majority of the experiments, except where noted otherwise. STED imaging was performed by J.Kompa. ExM was performed by D.S. MD simulation, peptide synthesis, and $K_d$ determination by FP assay was jointly conducted by R.M. and A.A. The generation of CRISPR/Cas9 KI cell lines were done by Y-H.L., B.K., and P.B. K.H., J.W., and Y-H.L. jointly carried out Hpep engineering experiments. Y-H.L., S.K., and T.M. jointly

conducted yeast display selection. NGS data analysis and in vitro characterization work was done by Y-H.L. with supports from J.W. Hpep-EGFP fusion characterization was performed jointly by Y-H.L. and N.F. The manuscript was written by Y-H.L and K.J., with contributions and feedback from all authors.

## Funding

## Competing interests
J.W., L.N., Y.-H.L., J.H., and K.J. are inventors of a patent with the title "Improved Split-HaloTags" filed by the Max Planck Society. The remaining authors declare no competing interests.
