## [Transparent Peer Review file · Nature Communications]

A high-affinity split-HaloTag for live-cell protein labeling

Corresponding Author: Professor Kai Johnsson

Version 0:

Reviewer comments:

Reviewer #1

(Remarks to the Author)

The work by Lin et al (“A high-affinity split-HaloTag for live-cell protein labeling”) is important for the field, of high impact and a thorough demonstration of developing split-HaloTag variants, as well as demonstrating a series of applications for different imaging applications. Especially the versatility of this tools for live imaging applications is important.

I only have minor comments.

When developing and characterizing the split-HaloTag variants, the authors use a clever yeast display approach. However, I recommend making sure to use precise language around describing “affinity” / “EC50” / “binding” / “kinetics”. In some places (for example the Extended Figure 2 legend), the authors (correctly) describe the EC50 measurements as a “proxy for binding affinity”. But, the development of the split fragments is advertised as engineering “high affinity” pairs, without clarifying that they are measuring a “proxy” for affinity in the main text. The use of EC50 as a “proxy” should be mentioned in the main text early on. The EC50 measurement is introduced very briefly and it is phrased somewhat confusing (on page 5), mentioning “labeling speed” and affinity, when neither direct kinetic or thermodynamic measurements are done, as one could perhaps expect when reading about “speed” and affinity.

In Figure 2, I recommend indicating clearly that the cartoon (a and b) illustrates the yeast display, whereas all other panels (as far as I can tell) report on characterization of purified proteins / peptides.

In the section about biophysical characterization (around line 100), I recommend checking the language. The authors should clarify precisely what the EC50 values report on, and how the dissociating constant K_d is measured (by a different technique). Around line 130, a complimentary biophysical technique is used that supports the apparent K_d values from the EC50 measurements, this should be more clearly highlighted. This is phrased precisely in reference 54 (now a Protein Science publication).

For cell imaging data presentation: A statement should be added to the legend of all figures that include cell images that reports on how many times this observation was made (ideally, how many independent experiments, and how many cells). This is done only in Figure 4 (“ $n=20$ microtubule filaments, $n>2$ ”). A description to indicate what was used as the basis for the “representative image” should be done for all cell images.

I recommend displaying microscopy images as black and white as much as possible. For example, red on black images are very hard to see.

The authors comment on comparisons of signal-to-background in quantitative terms (“higher”, line 158, “moderate reduction”, line 175, “well-correlated” intensities, line 185). It could be convincing to quantify this, or at least make this effect of comparing imaging results more visible by using black and white images.

The laser intensities in the methods is listed as a % value – this is very hard to reproduce, I recommend measuring the laser intensity directly and reporting that.

In line 169 an experiment in HeLa cells is mentioned – but in the discussion, only U2OS and HEK cells are talked about as examples for cell lines used.

In Figure 5: please add scale bars to the images.

Reviewer #2

(Remarks to the Author)

Yin-Hsi Lin and coworkers describe a new high-affinity split-HaloTag system for imaging cellular proteins. Overall, this is a strong paper. They report noteworthy results and rigorous data. The presented work will be of interest to people in the field of protein tag development and, more broadly, fluorescence imaging of cells. The work supports the claims with rigorous validation of the new tags. I will describe two strengths of the paper. First, direct measurement of the affinity of the new

peptide tags for cpHaloTag by more than 1 method. Two, excellent compatibility of the new tags with expansion microscopy, FLIM, and STED. The methods are sufficiently detailed to be reproduced.

I recommend publication with only suggested minor revisions. These can be made at the authors' discretion.

- 1) P. 3, Line 27: The authors conclude that work cited in (11, 13-25) all suffer from low signal-to-background ratio. I don't think that this is accurate. The papers cited should be a smaller subset of this list. Some of those papers have data with excellent SBR.
- 2) In many places, the fluorescent ligand (e.g., CA-CPY, CA-MaP555, or CA-SiR) is referred to with acronyms. This becomes confusing. Perhaps some of the ligands could just have the full name? (i.e., CA-silicon-rhodamine)
- 3) The introduction should include more on existing split-HaloTags. For example, the work described in citations 45-48 should be mentioned in the introduction.
- 4) The section titled "Hpep11 allows cloning-free CRISPR-Cas9 genome editing" should include citations for the method used, such as the Leonetti papers (e.g., citations 12, 28, 29). As written, it is not clear that the current work leveraged knock-in methods described by the team at UCSF.
- 5) Figure 4 compares CLSM with STED. It also compares HaloTag with Split-HaloTag. It is hard to tell from the line plots that the signal from HaloTag is brighter than for the new Split-HaloTag. Perhaps the plots could be shown with intensity values instead of normalized? My concern is that the claim on p. 14 (lines 274-276) that the signal is comparable is not quite accurate, and that it is more "lower but close to comparable".
- 6) The actual diameter of microtubules should be stated and cited (p. 14, line 283).

Reviewer #3

(Remarks to the Author)

The manuscript by Lin et al. is a strong and impressive new technology that describes the design of a high-affinity split HaloTag system for live-cell protein labeling. The design is modular, based on a short peptide fragment that is amenable for cellular tagging and versatile to maintain multifunctional ligands to react with HaloTag after attachment, localizing HaloTag to the protein of interest. Prior to publication, there are some important revisions for the work that will strengthen the narrative:

1. Adding a table that summarizes and compares the direct performance of the Hpep variants would be helpful for the discussion, providing contrast between the measured K_d, EC₅₀, etc. across the different experiments with discussion for potential variances between the results.
2. In the manuscript, you provide discussion about the advantages to this system over similar systems (such as ALFA-tag). Have these been compared directly using your system under the same cellular conditions?
3. Why was the peptide fragment designed to be a high-affinity binder, rather than building a covalent split-protein that rejoins (such as a split-GFP)? Was it just that the yeast library did not yield covalent systems?
4. The discussion section provided strong support for the new strategy, but it would be helpful to add discussion related to other topics that broaden the impact of the work – which, I will commend is highly impressive. Some of these related discussion points center around different questions I have when reading the manuscript:
 - i. Are there concerns for potential immunogenicity and toxicity of the system if advanced to in vivo? What are potential applications of the technology in biosensor design for various cellular systems?
 - ii. If this technology was applied for drug delivery, there are some great ways that I could see it being used to help localize at diseased tissue sites – what are ways that this technology could resolve some of the existing gaps in this field?
 - iii. Some of these alternative strategies for protein tagging (such as MoonTag) allow for multi-color imaging with a simple molecule. Can this technology be scaled/multiplexed up to similar levels?
 - iv. What are the limitations that you see in this technology currently?

Version 1:

Reviewer comments:

Reviewer #1

(Remarks to the Author)

The authors have adequately addressed my comments and concerns in the revised manuscript.

Reviewer #2

(Remarks to the Author)

The authors have done a good job addressing reviewer comments. I recommend publication without further revision.

Reviewer #3

(Remarks to the Author)

This manuscript is acceptable for publication.

REVIEWER COMMENTS

Reviewer #1 (Remarks to the Author):

The work by Lin et al (“A high-affinity split-HaloTag for live-cell protein labeling”) is important for the field, of high impact and a thorough demonstration of developing split-HaloTag variants, as well as demonstrating a series of applications for different imaging applications. Especially the versatility of this tools for live imaging applications is important. I only have minor comments.

We thank the reviewer for the positive and supportive comments. The manuscript has been revised as described below.

Specific Comments:

When developing and characterizing the split-HaloTag variants, the authors use a clever yeast display approach. However, I recommend making sure to use precise language around describing “affinity” / “EC50” / “binding” / “kinetics”. In some places (for example the Extended Figure 2 legend), the authors (correctly) describe the EC50 measurements as a “proxy for binding affinity”. But, the development of the split fragments is advertised as engineering “high affinity” pairs, without clarifying that they are measuring a “proxy” for affinity in the main text. The use of EC50 as a “proxy” should be mentioned in the main text early on. The EC50 measurement is introduced very briefly and it is phrased somewhat confusing (on page 5), mentioning “labeling speed” and affinity, when neither direct kinetic or thermodynamic measurements are done, as one could perhaps expect when reading about “speed” and affinity.

We thank the reviewer for this suggestion to use more precise terminology to avoid potential confusion. Accordingly, we have revised the manuscript by replacing the terms “binding affinity” and “labeling efficiency” with more appropriate alternatives when no direct thermodynamic or kinetics measurements were performed.

In Figure 2, I recommend indicating clearly that the cartoon (a and b) illustrates the yeast display, whereas all other panels (as far as I can tell) report on characterization of purified proteins / peptides.

In the section about biophysical characterization (around line 100), I recommend checking the language. The authors should clarify precisely what the EC50 values report on, and how the dissociating constant K_d is measured (by a different technique). Around line 130, a complimentary biophysical technique is used that supports the apparent k_D values from the EC50 measurements, this should be more clearly highlighted. This is phrased precisely in reference 54 (now a Protein Science publication).

- In Figure 1, panel (a) illustrates the overall concept of using Hpep as a labeling tool, and panel (b) depicts the yeast display screening workflow for cpHaloΔ engineering, as clarified in the figure legend. In addition, we have added the following sentence to specify the content of the remaining panels: “(c-g), *in vitro* characterization of purified split-HaloTag fragments.”
- To highlight the distinction between the affinity measurement techniques, we have revised the “*Biophysical characterization of split-HaloTag pairs*” section and added the following introductory sentence:
“To directly assess the binding interaction between Hpep and cpHaloΔ, biophysical measurements were performed to complement the labeling reaction-based EC₅₀ analyses described above.”

For cell imaging data presentation: A statement should be added to the legend of all figures that include cell images that reports on how many times this observation was made (ideally, how many independent experiments, and how many cells). This is done only in Figure 4 (“n=20 microtubule filaments, n>2”). A description to indicate what was used as the basis for the “representative image” should be done for all cell images.

We thank the reviewer for this helpful suggestion. We have now included in the legend of each representative image the number of independent biological experiments performed and the number of cells or fields of view analyzed.

I recommend displaying microscopy images as black and white as much as possible. For example, red on black images are very hard to see. The authors comment on comparisons of signal-to-background in quantitative terms (“higher”, line 158, “moderate reduction”, line 175, “well-correlated” intensities, line 185). It could be convincing to quantify this, or at least make this effect of comparing imaging results more visible by using black and white images.

We thank the reviewer for the helpful suggestion. In response, we have provided the grayscale versions of the fluorescence images in addition to the original pseudocolor panels for better visualization.

- The Lamin B1 labeling images in Figure 2a and 2b are represented in grayscale in Supplementary Fig4 with intensity profiles shown to highlight the “higher” SBR achieved by Hpep11 compared to Hpep9 in live-cell protein labeling.
- “moderate reduction”, line 175 => The reduction in EGFP signal refers to the flow cytometry data represented in Extended Data Fig 7a-b, but not referring to the imaging data shown in Extended Data Fig 7c. It is clarified in the corresponding highlighted text (line 214, page 10).
- A merged image of the CP-TMR (SNAPf–Hpep11) and CA-SiR (cpHaloΔ3) channels is now provided in Supplementary Fig. 4e, and the corresponding overlaid intensity profiles are shown in Supplementary Fig. 4f, confirming the strong “correlation” between the fluorescence signals in both channels.

The laser intensities in the methods is listed as a % value – this is very hard to reproduce, I recommend measuring the laser intensity directly and reporting that. In line 169 an experiment in HeLa cells is mentioned – but in the discussion, only U2OS and HEK cells are talked about as examples for cell lines used. In Figure 5: please add scale bars to the images.

We thank the reviewer for these helpful suggestions. We have made the following revisions accordingly:

- We have included the laser intensities of our confocal and STED microscope in two supplementary tables (Supplementary Table 19 and Supplementary Table 21) to facilitate reproducibility.
- We have revised the manuscript to include all three cell lines (U2OS, 293FT, and HeLa) and clarified that HeLa cells were also used to demonstrate the feasibility of the Hpep tagging method.
- We originally mentioned only U2OS and 293FT cells since we wanted to highlight the cell lines we have tested for CRISPR/Cas9 KI, but we have included in the revised manuscript all the three cell lines including HeLa cells in which we have shown the feasibility of Hpep tagging method.
- We have added scale bars to all images shown in Figure 5.

Reviewer #2 (Remarks to the Author):

Yin-Hsi Lin and coworkers describe a new high-affinity split-HaloTag system for imaging cellular proteins. Overall, this is a strong paper. They report noteworthy results and rigorous data. The presented work will be of interest to people in the field of protein tag development and, more broadly, fluorescence imaging of cells. The work supports the claims with rigorous validation of the new tags. I will describe two strengths of the paper. First, direct measurement of the affinity of the new peptide tags for cpHaloTag by more than 1 method. Two, excellent compatibility of the new tags with expansion microscopy, FLIM, and STED. The methods are sufficiently detailed to be reproduced.

I recommend publication with only suggested minor revisions. These can be made at the authors' discretion.

We appreciate the encouraging feedbacks from the reviewer and revised the manuscript according to the reviewer's suggestions.

1) P. 3, Line 27: The authors conclude that work cited in (11, 13-25) all suffer from low signal-to-background ratio. I don't think that this is accurate. The papers cited should be a smaller subset of this list. Some of those papers have data with excellent SBR.

We thank the reviewer for this comment and agree that not all studies cited in (11,13-25) report low SBR. Our intention was not to imply that all these systems inherently yield low SBR, but rather to highlight that their application to low-abundance, endogenously expressed proteins remains limited, likely due to the difficulty of achieving high SBR in such contexts.

To clarify this point, we have revised the corresponding sentence in the main text as follows:

“However, their use in live-cell imaging of intracellular proteins remains limited due to (i) cytotoxicity^{13,14,24,25}, (ii) low cell permeability of the substrate/probe^{17-20,23-25}, and/or (iii) challenges in achieving a sufficiently high signal-to-background ratio (SBR). Although some of these systems perform well for overexpressed proteins^{11,13-25}, for applications to low-abundance, endogenously expressed proteins, the SBR remains a key challenge when only a single copy of the tag is fused.

This revised text clarifies that while some systems provide excellent SBR for overexpressed proteins, achieving high SBR for endogenously expressed targets remains a challenge.

2) In many places, the fluorescent ligand (e.g., CA-CPY, CA-MaP555, or CA-SiR) is referred to with acronyms. This becomes confusing. Perhaps some of the ligands could just have the full name? (i.e., CA-silicon-rhodamine)

We thank the reviewer for this helpful suggestion. To improve clarity, we have made sure to include the full name of each fluorescent ligand at its first appearance in the article. In addition, a supplementary table (Supplementary Table 6) has been added that lists all fluorescent ligands used in this study, together with their full chemical names, well-recognized alternative names, and/or commercial names. We have retained the commonly used acronyms throughout the rest of the manuscript for better readability, as these abbreviations are well established and widely recognized in the fluorescence imaging field.

3) The introduction should include more on existing split-HaloTags. For example, the work described in citations 45-48 should be mentioned in the introduction.

We thank the reviewer for this suggestion. We have added sentences in the introduction to cite the existing split-HaloTag systems.

4) The section titled "Hpep11 allows cloning-free CRISPR-Cas9 genome editing" should include citations for the method used, such as the Leonetti papers (e.g., citations 12, 28, 29). As written, it is not clear that the current work leveraged knock-in methods described by the team at UCSF.

We thank the reviewer for this suggestion. We have added the phrase "Guided by insights from previous studies" and included the relevant citations to clarify that the design of our endogenous tagging experiments leveraged the knock-in strategies previously described by the UCSF team.

5) Figure 4 compares CLSM with STED. It also compares HaloTag with Split-HaloTag. It is hard to tell from the line plots that the signal from HaloTag is brighter than for the new Split-HaloTag. Perhaps the plots could be shown with intensity values instead of normalized? My concern is that the claim on p. 14 (lines 274-276) that the signal is comparable is not quite accurate, and that it is more "lower but close to comparable".

We thank the reviewer for this insightful comment. To address this concern, we have added a new supplementary figure (Supplementary Fig. 5) presenting a side-by-side comparison of HaloTag and split-HaloTag in live-cell imaging of endogenously tagged TOM20. This figure includes representative CLSM and STED images acquired under identical labeling and imaging conditions, enabling a direct visual comparison of signal intensities and SBR between the two systems. The data confirm that split-HaloTag provides an overall comparable performance to HaloTag in this context.

6) The actual diameter of microtubules should be stated and cited (p. 14, line 283).

We thank the reviewer for this helpful suggestion. We have now included a reference to previously reported measurements of microtubule dimensions for context.

Reviewer #3 (Remarks to the Author):

The manuscript by Lin et al. is a strong and impressive new technology that describes the design of a high-affinity split HaloTag system for live-cell protein labeling. The design is modular, based on a short peptide fragment that is amenable for cellular tagging and versatile to maintain multifunctional ligands to react with HaloTag after attachment, localizing HaloTag to the protein of interest. Prior to publication, there are some important revisions for the work that will strengthen the narrative:

We thank to the reviewer's positive comments and the insightful suggestions for further strengthening the manuscript.

1. Adding a table that summarizes and compares the direct performance of the Hpep variants would be helpful for the discussion, providing contrast between the measured Kd, EC50, etc. across the different experiments with discussion for potential variances between the results.

We thank the reviewer for this suggestion. We have added a summary table (Supplementary Table 22) listing the key binding properties (Kd, EC50, and fluorescence turn-on) of the Hpep variants in complex with cpHalo Δ 3 across different assays. Additionally, we provide a brief discussion highlighting potential sources of variation, including differences in assay format (e.g., solution-based versus immobilized), Hpep modification (fluorophore or biotin conjugation), and accessibility effects. This table allows direct comparison of the Hpep variants and supports the discussion of their relative performance.

2. In the manuscript, you provide discussion about the advantages to this system over similar systems (such as ALFA-tag). Have these been compared directly using your system under the same cellular conditions?

We thank the reviewer for this comment. Our discussion focuses on conceptual differences between our system and other tags, such as ALFA-tag. While we have not performed direct side-by-side comparisons, the approaches are complementary and each offers advantages depending on the application.

The ALFA-tag is effective for applications such as affinity purification and western blotting, with high affinity and compatibility with fixation. However, live-cell imaging of low-abundance proteins can be challenging due to background from unbound fluorescent NbALFA, often requiring multiple tag copies or careful expression control. Multiplexing has not been demonstrated.

In contrast, the split-HaloTag system provides high signal-to-background without optimizing cpHalo Δ 3 expression, is well-suited for imaging endogenously tagged

proteins with super-resolution microscopy, and allows multiplexing with multiple Hpep variants using a single cpHalo Δ .

Thus, we believe direct comparisons under all conditions are not necessary, as the optimal system depends on the experimental context.

3. Why was the peptide fragment designed to be a high-affinity binder, rather than building a covalent split-protein that rejoins (such as a split-GFP)? Was it just that the yeast library did not yield covalent systems?

We thank the reviewer for this insightful comment. Split-FP, such split-GFP, are not covalently reconstituting systems; rather, their complementation relies on tight noncovalent interactions between the two split fragments. For the engineering of split-HaloTag, we aimed to achieve similarly strong, spontaneous association of the two fragments. From our two cpHalo Δ library screening campaigns, we identified variants that can be efficiently labeled at low nanomolar Hpep concentrations. Further reduction in Hpep concentrations in the screening experiments did not result in detectable labelling under flow cytometry, suggesting that we reached the detection sensitivity limit in the yeast display experiments. Having achieved our target EC50 (single-digit nanomolar), we proceeded to validate and optimize the variants in vitro and in cells without additional rounds of library design and screen. While engineering splitHaloTagpairs with even higher binding affinity might be possible, it may hinder some applications requiring reversible binding.

4. The discussion section provided strong support for the new strategy, but it would be helpful to add discussion related to other topics that broaden the impact of the work – which, I will commend is highly impressive. Some of these related discussion points center around different questions I have when reading the manuscript:

i. Are there concerns for potential immunogenicity and toxicity of the system if advanced to in vivo? What are potential applications of the technology in biosensor design for various cellular systems?

We thank the reviewer for raising this point. Various HaloTag-derived tools, including systems incorporating similar Hpep sequences (*Huppertz, M. C. et al., Science, 2024*) have been successfully applied in vivo (e.g. in mice, fish and plants) without reported issues of immunogenicity and toxicity. This suggests that our split-HaloTag system is also likely to be well-tolerated in such contexts. We envision that the system could be adapted for biosensor designs or in vivo applications. However, we note that the bioavailability and pharmacokinetics of the currently available HaloTag ligands remain important factors that would need further optimization for efficient in vivo implementation.

We have revised the discussion section to highlight the potential application of our system in more complex organisms, including in vivo, where various HaloTag-derived tools have been successfully employed. The relevant references have been added accordingly.

ii. If this technology was applied for drug delivery, there are some great ways that I could see it being used to help localize at diseased tissue sites – what are ways that this technology could resolve some of the existing gaps in this field?

We appreciate the reviewer's very interesting suggestion. While drug delivery applications are beyond the scope of the current work, the modularity and high specificity of the split-HaloTag system could, in principle, be adapted for various purposes, including targeted delivery or controlled activation strategies. For example, therapeutic or bioactive moieties could be conjugated to HaloTag ligands or Hpep to help them specifically localize to the desired sites (with site-specific expression of the other complementary fragment) through the affinity between split-HaloTag fragments. Nevertheless, realizing such applications would require careful investigation and optimization of factors such as the bioavailability of the conjugates and their potential impact on the affinity of the split-HaloTag interaction.

iii. Some of these alternative strategies for protein tagging (such as MoonTag) allow for multi-color imaging with a simple molecule. Can this technology be scaled/multiplexed up to similar levels?

We thank the reviewer for raising this point. As discussed in the introduction and discussion sections, MoonTag requires co-expression of a nanobody-FP fusion for live-cell imaging. In this study, we demonstrated that multiplexed labeling is feasible with our split-HaloTag system. Two targets can be imaged simultaneously by combining distinct Hpep variants with a single cpHalo Δ 3 using fluorescence lifetime imaging. While MoonTag can also be used together with other peptide-tagging systems (e.g., SunTag) for multiplexed imaging, each peptide-tag requires co-expression of its corresponding nanobody-FP fusion, making such an approach increasingly complex as the number of targets grow. This point is highlighted in the discussion section with split-FP systems.

In addition, MoonTag supports signal amplification through tandem peptide arrays. Technically, such a strategy can be extended to our Hpep system. However, in the preliminary experiments, constructs containing 5x or 10x Hpep repeats fused to TOM20 or LaminB1 did not yield stronger labeling signal, analyzed through live-cell imaging. In the case of TOM20, the 10x Hpep11 repeat caused aggregation, likely due to charge interactions between Hpep repeats or with other cellular components. These observations indicate that array-based signal amplification would require further optimization (e.g., linker design), which is beyond the scope of the current study.

iv. What are the limitations that you see in this technology currently?

We thank the reviewer for raising this point. In response, we have revised the Discussion to more clearly articulate both the advantages and limitations of our system. Specifically, we insert a comment on what we see as the main limitation, which is the requirement for co-expression of an appropriately localized cpHalo Δ .